# European topsoil bulk density and organic carbon stock database (0-20 cm) using machine learning based pedotransfer functions

Songchao Chen[1,2#], Zhongxing Chen[1,2#], Xianglin Zhang[1,2,3], Zhongkui Luo[2], Calogero Schillaci[4], Dominique Arrouays[5], Anne C. Richer-de-Forges[5], Zhou Shi[2]

[1]ZJU-Hangzhou Global Scientific and Technological Innovation Center, Zhejiang University, Hangzhou, 311215, China

[2]College of Environmental and Resource Sciences, Zhejiang University, Hangzhou, 310058, China

[3]UMR ECOSYS, AgroParisTech, INRAE, Université Paris-Saclay, Palaiseau 91120, France

[4]European Commission, Joint Research Centre, Ispra, 21026, Italy

[5]INRAE, Info&Sols, Orléans, 45075, France

# These authors contributed equally.

*Correspondence to*: Zhou Shi (Email: shizhou@zju.du.cn)

**Abstract.** Soil bulk density (BD) serves as a fundamental indicator of soil health and quality, exerting a significant influence on critical factors such as plant growth, nutrient availability, and water retention. Due to its limited availability in soil databases, the application of pedotransfer functions (PTFs) has emerged as a potent tool for predicting BD using other easily measurable soil properties, while the impact of these PTFs' performance on soil organic carbon (SOC) stock calculation has been rarely explored. In this study, we proposed an innovative local modelling approach for predicting BD of fine earth ($BD_{fine}$) across Europe using the recently released $BD_{fine}$ data from the LUCAS Soil 2018 (0-20 cm) and relevant predictors. Our approach involved a combination of neighbour sample search, Forward Recursive Feature Selection (FRFS) and Random Forest (RF) model (local-$RF_{FRFS}$). The results showed that local-$RF_{FRFS}$ had a good performance in predicting $BD_{fine}$ ($R^2$ of 0.58, root mean square error (RMSE) of 0.19 g cm$^{-3}$, relative error (RE) of 16.27%), surpassing the earlier published PTFs ($R^2$ of 0.40-0.45, RMSE of 0.22 g cm$^{-3}$, RE of 19.11-21.18%) and global PTFs using RF with and without FRFS ($R^2$ of 0.56-0.57, RMSE of 0.19 g cm$^{-3}$, RE of 16.47-16.74%). Interestingly, we found the best earlier published PTF ($R^2$=0.84, RMSE=1.39 kg m$^{-2}$, RE of 17.57%) performed close to the local-$RF_{FRFS}$ ($R^2$=0.85, RMSE=1.32 kg m$^{-2}$, RE of 15.01%) in SOC stock calculation using $BD_{fine}$ predictions. However, the local-$RF_{FRFS}$ still performed better ($\Delta R^2$>0.2) for soil samples with low SOC stock (<3 kg m$^{-2}$). Therefore, we suggest that the local-$RF_{FRFS}$ is a promising method for $BD_{fine}$ prediction while earlier published PTFs would be more efficient when $BD_{fine}$ is subsequently utilized for calculating SOC stock. Finally, we produced two topsoil $BD_{fine}$ and SOC stocks datasets (18,945 and 15,389 soil samples) at 0-20 cm for LUCAS Soil 2018 using the best earlier published PTF and local-$RF_{FRFS}$, respectively. This dataset is archived from the Zenodo platform at https://zenodo.org/records/10211884 (Chen et al., 2023a).The outcomes of this study present a meaningful advancement in enhancing the predictive accuracy of

BD$_{fine}$, and the resultant BD$_{fine}$ and SOC stock datasets for topsoil across the Europe enable more precise soil hydrological and biological modelling.

## 1 Introduction

Soil plays a pivotal role in supporting ecosystems and sustaining life on our planet (Rabot et al., 2018). Its physical properties are crucial for various disciplines such as agriculture, environmental science, and land management. Among these properties, soil bulk density (BD) holds particular significance as it serves as a fundamental indicator of soil health, structure, and water holding capacity. BD directly influences vital factors like plant growth, nutrient availability, and overall soil quality (Dam et al., 2005; Chen et al., 2018; Schillaci et al., 2021). Additionally, BD plays a crucial role in computing stock of water, chemical elements (e.g., soil organic carbon, SOC) or compounds by soil surface unit or soil volume unit, making it even more essential in soil studies. Nonetheless, the uncertainty in SOC stock estimates arises due to the variations in methods used to substitute for missing BD data (Benites et al., 2007; Dawson and Smith, 2007; Wiesmeier et al., 2012; Chen et al., 2023b). It is important to acknowledge that BD in topsoil exhibits considerable variations across different geographical regions due to factors like diverse soil types, climate conditions, vegetation cover, and land cover patterns (Hollis et al., 2012; Lark et al., 2014; Li et al., 2019). These regional disparities underscore the need for a comprehensive understanding of BD in soil research and its implications for various aspects of ecosystem functioning and management.

Characterizing the spatial distribution of BD across a diverse and extensive continent like Europe presents a complex challenge (Chen et al., 2018; Nasta et al., 2020; Palladino et al., 2022; Panagos et al., 2024). Conventional soil sampling and laboratory analyses are time-consuming, costly, and impractical at a broad scale (Makovníková et al., 2017). In response to this challenge, the development of pedotransfer functions (PTFs) has emerged as a powerful approach (Van Looy et al., 2017). PTFs are mathematical models that estimate soil properties, such as BD, based on readily available and easily measurable soil data (e.g., SOC, clay, silt, and sand). These functions serve as invaluable tools for predicting soil properties at unvisited locations, facilitating regional soil mapping, and enhancing our understanding of soil dynamics across vast areas (Chen et al., 2018; Schillaci et al., 2021; Palladino et al., 2022). Furthermore, the incorporation of globally available predictor variables, such as topography and land cover, showed a promise in enhancing the effectiveness and applicability of PTFs for gap-filling of BD data (Ramcharan et al., 2017; Bondi et al., 2018; Patton et al., 2019).

In the early stage, PTFs predominantly employed regression techniques due to their simplicity (Gupta and Larson, 1979; Rawls and Brakensiek, 1985). However, with advancements in science and technology, a wide range of models have been developed for deriving PTFs, particularly for continuous predicted variables. These methods encompass linear regression, generalized linear models, generalized additive models, regression trees, artificial neural networks, support vector machines, gradient boosted model, and random forests (RF) (Van Looy et al., 2017; Chen et al., 2018). The utilization of these advanced techniques has substantially improved the accuracy of BD prediction (Table 1).

**Table 1** Summary of previous studies on using PTFs for BD prediction across scales. $R^2$ indicates the determination coefficient.

| ID | Scale | Sample size | Model | $R^2$ | Reference |
|---|---|---|---|---|---|
| 1 | Landscape | 164 | Naive-BN | 0.26 | Taalab et al. (2015) |
| | | | Hierarchical-BN | 0.42 | |
| 2 | National | 2,462 | MLR | 0.41 | Katuwal et al. (2020) |
| | | | RF | 0.62 | |
| | | | RR | 0.60 | |
| | | | ANN | 0.61 | |
| 3 | National | 1,357 | GBM | 0.53 | Chen et al. (2018) |
| 4 | Regional | 169 | k-NN | 0.32 | Ghehi et al. (2012) |
| | | | BRT | 0.30 | |
| 5 | National | 485 | GBM | 0.67 | Jalabert et al. (2010) |
| 6 | Regional | 495 | ANN | 0.71 | Yi et al. (2016) |
| | | | MLR | 0.63 | |
| 7 | National | 188 | MLR | 0.21 | Schillaci et al. (2021) |
| | | | MLR-BS | 0.38 | |
| | | | ANN | 0.48 | |

MLR, multiple linear regression; RF, random forest; RR, regression rules; ANN, artificial neural networks; GBM, generalized boosted models; BRT, boosted regression trees; BN, Bayesian network; k-NN, k-nearest neighbour; MLR-BS, multiple linear regression (stepwise variable selection)

PTFs have emerged as an alternative approach to address the scarcity of BD data (Van Looy et al., 2017). They have been implemented and tested in diverse regions and countries, providing a practical and cost-effective means for predicting BD using readily available soil properties. However, it is noteworthy that the majority of previous studies utilizing machine learning (ML) and PTFs for BD prediction have been conducted at regional or national scales, with limited research focusing on the intercontinental scale (Taalab et al., 2015; Shiri et al., 2017; Katuwal et al., 2020). Despite the accomplishments of PTFs in BD estimation, a gap emerges when transitioning to global modelling (a fixed model to predict all the unknown samples) endeavours. The reliance on global models, while useful in capturing broad patterns, often faces constraints in delivering accurate predictions at finer scales (Gupta et al., 2018). These global models may fail to account for the nuanced spatial and environmental variations that play a pivotal role in determining BD across different landscapes. While numerous studies have harnessed ML based PTFs (ML-PTFs) to improve the model performance for BD at national and regional levels, the expansion of these methodologies to encompass continental contexts remains relatively limited (Nasta et al., 2020; Schillaci et al., 2021; Palladino et al., 2022). This gap underscores the need for a modelling approach that bridges the gap between broad-scale global modelling and context-specific requirements of diverse regions and ecosystems (Wang et al., 2024). This is where the concept of local modelling steps in. The local model adopts a dynamic modelling strategy: it firstly selects a part of similar samples close to each unknown sample in the predictor space, then it fits a predictive model using the selected similar samples (not the whole data). Since the selected similar samples vary for each unknown sample, the corresponding

local model is different from others. Local modelling strategy enables the consideration of environmental relevance by clustering data under similar environmental conditions (i.e. in the present case similar predictors feature space, including soil

properties, elevation, land cover and climate conditions), which aids in constructing specialized PTFs that capture soil-environment relationships (Nocita et al., 2014; Chen et al., 2018). Thus, there is a compelling need for further investigations and developments in local modelling techniques to improving BD predictions. Furthermore, despite of the widely use of PTFs for predicting BD in SOC stock calculation from continental to global scales, how the performances (e.g., $R^2$, root mean square error, relative error) of PTFs based BD prediction impact the quality of SOC stock remains poorly explored (Cotrufo et al.,

2019; Augusto and Boč, 2022; Wang et al., 2022; De Rosa et al., 2023).

To address aforementioned issues, we investigated RF model in combination with variable selection and local modelling strategy, to evaluate the potential of different PTFs in BD prediction as well as SOC stock calculation. The main objectives of this study are as follows:

(1) to compare the performances of earlier published PTFs and ML-PTFs for BD prediction;

(2) to evaluate the potential of local modelling strategies for BD prediction;

(3) to investigate the impact of PTFs-based BD prediction on the accuracy of SOC stocks calculation.

## 2 Materials and methods

### 2.1 Soil data

The soil data were compiled from the Land Use and Coverage Area Frame Survey Soil (LUCAS Soil) campaigns conducted

in 2009, 2015 and 2018 (Fernández-Ugalde et al., 2022; Panagos et al., 2022). The survey encompassed a stratified random sampling approach, which identified approximately 20,000 topsoil sampling locations across the European Union (EU) and the United Kingdom (UK) for each campaign. At each sampling site (circle of 4 m diameter plot), 5 topsoil samples (0–20 cm) were collected after the removal of the litter layer, and the land cover (LC) was recorded. These samples were then combined into a bulked composite topsoil sample for analysis. Subsequently, all topsoil samples underwent air-drying and sieving to less

than 2 mm. Standard laboratory analysis was conducted in an accredited laboratory (Kecskemét, Hungary), including particle size fractions (clay content, silt content, sand content, %), coarse fragments (mass fraction, %/100), BD (whole mass, g cm$^{-3}$), pH (in water), SOC content (g kg$^{-1}$), carbonates (CaCO$_3$, g kg$^{-1}$), total nitrogen (N, g kg$^{-1}$), extractable potassium (K, mg kg$^{-1}$), cation exchange capacity (CEC, cmol(+) kg$^{-1}$). For more comprehensive information about LUCAS Soil 2009/2015/2018, we refer to Orgiazzi et al. (2022). In the LUCAS Soil 2018 survey, topsoil sampling was conducted across all EU Member States

and UK, employing the identical set of 25,947 locations that were targeted during the 2015 survey (Fernández-Ugalde et al., 2022). However, due to the absence of particle-size fractions in LUCAS Soil 2018, we resorted to use the data from LUCAS Soil 2009/2015 by the unique identifier soil ID (Panagos et al., 2022). To ensure the reliability of the data, we excluded samples with soil particle fractions recorded as 0. Finally, 5,163 topsoil samples were retained for further analysis (Fig. 1). In the

following parts of the article, we define $BD_{sample}$ as the whole soil mass:volume ratio, and $BD_{fine}$ as the fine earth mass:volume ratio.

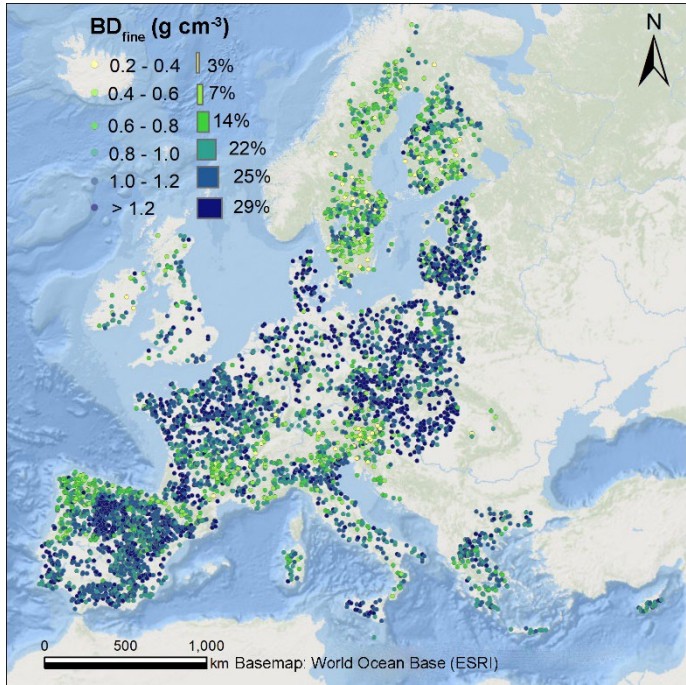

**Figure 1** Spatial distribution of 5,163 topsoil samples with estimated $BD_{fine}$ from LUCAS Soil 2018. The colors represent six $BD_{fine}$ levels and the histogram represents the relevant percentages for these $BD_{fine}$ levels.

Since $BD_{sample}$ was measured for the whole mass and CF was measured as a mass fraction ($CF_{massfraction}$) in part of the topsoil samples of the LUCAS Soil 2015/2018, they cannot be used directly to accurately calculate SOC stock. Note that if the mass of fine fraction has been measured and recorded, as the total volume of the sample is known, the SOC stock can be calculated directly (Poeplau et al., 2017). However, in numerous locations, neither the mass of fine fraction nor $BD_{sample}$ were measured. This is why we needed to estimate and use $BD_{fine}$ and $CF_{volumefraction}$ in order to calculate SOC stocks where $BD_{sample}$ was missing (Poeplau et al., 2017). To this aim, we used a recently released dataset for $BD_{fine}$ and $CF_{volumefraction}$ by Pacini et al. (2023) based on $BD_{sample}$ and $CF_{massfraction}$ from LUCAS Soil 2018.

**2.2 Predictor variables related to relief, climate, and land cover**

The elevation (ELE) was derived from Shuttle Radar Topography Mission (SRTM) 1-km Digital Elevation Model (Farr et al., 2007). Climatic data, including mean annual precipitation (MAP) and mean annual temperature (MAT), were acquired from the WorldClim Version 2 at 1 km resolution (Fick and Hijmans, 2017). The Global-PET dataset at 1 km resolution was used to extract potential evapotranspiration (PET) and aridity index (AI) (Zomer et al., 2022). The land cover (LC) was directly derived from the records of LUCAS Soil 2018 during soil sampling campaign.

## 2.3 Earlier published PTFs

We evaluated four earlier published PTFs that have been widely used to estimated $BD_{sample}$ or $BD_{fine}$ in previous studies at both local and broad scales (Adams, 1973; Atwood et al., 2017; Chen et al., 2018; Sun et al., 2020; Tao et al., 2023). For PTF-3 and PTF-4, soil organic matter (SOM) content was determined by the conversion factor of 1.724 using SOC content. In the present study, we used these PTFs to estimate $BD_{fine}$. The parameters in these PTFs were refitted by the Levenberg-Marquardt non-linear least-square method available in the minpack.lm R package based on our data (Bates and Watts, 1988; Zhu et al., 1997; Elzhov et al., 2015). These refitted parameters of PTFs are present in Table 2.

**Table 2** Summary of four earlier published PTFs defined in the literature.

| Model | Function | Refitted coefficients | | | References | $R^2$ |
|---|---|---|---|---|---|---|
| | | a | b | c | | |
| PTF-1 | $BD = a * \%SOC^b$ | 1.197 | -0.229 | / | Atwood et al. (2017) | 0.40 |
| PTF-2 | $BD = \dfrac{1}{a + b * \%SOC}$ | 0.733 | 0.0982 | / | Chen et al. (2018) | 0.45 |
| PTF-3 | $BD = \dfrac{100}{\dfrac{\%SOM}{0.244} - \dfrac{(100 - \%SOM)}{a}}$ | 1.231 | / | / | Adams (1973) Sun et al. (2020) | 0.41 |
| PTF-4 | $BD = a + b \times \exp(-c \times \%SOM)$ | 0.348 | 0.993 | 0.0882 | Tao et al. (2023) | 0.45 |

BD, bulk density; depending on authors cited in references, BD has been considered as BD of fine fraction ($BD_{fine}$) or BD of the whole sample ($BD_{sample}$), both expressed in g cm$^{-3}$; here, the refitted coefficients correspond to $BD_{fine}$; SOM, soil organic matter content (% in soil mass); SOC, soil organic carbon content (% in soil mass).

## 2.4 Global ML-PTFs

To compare with earlier published PTFs, we used random forest (RF) to construct global ML-PTFs for predicting $BD_{fine}$. RF is an ensemble learning method that aggregates predictions from multiple decision trees to obtain the final estimates of the target variable. In growing a decision tree, a random subsample of data is selected from the verification dataset, and a set of random predictor variables is used for splitting the subsampled data. Two parameters, ntree and mtry, were optimized by 10-fold cross-validation. Here, 15 predictor variables, such as sand content, silt content, clay content, SOC content, ELE (Table 3), were used to build the global RF model.

**Table 3** The variables used in RF$_{Full}$ and RF$_{FRFS}$. For RF$_{FRFS}$, the order of variables is listed by the descending importance. RF$_{Full}$ uses all potential predictors, even if they may be redundant of multi-collinear (typical case of the use of clay, silt and sand contents together). RF$_{FRFS}$ applies FRFS thus eliminating both multi-collinearity and irrelevant predictor variables (e.g., one particle size fraction (sand content) is left out). The abbreviations are detailed below: SOC, soil organic carbon content; CEC, cation exchange capacity; AI, aridity index; PET, potential evapotranspiration; MAP, mean annual precipitation; MAT,

mean annual temperature; ELE, elevation; LC, land cover. Clay, silt, sand, and $CaCO_3$ are expressed in %; pH is pH in a 1:2.5 soil:water mixture.

| Model | Selected predictors | Number of predictors |
|---|---|---|
| $RF_{Full}$ | clay, silt, sand, pH, SOC, $CaCO_3$, N, K, CEC, AI, PET, MAP, MAT, ELE, LC | 15 |
| $RF_{FRFS}$ | SOC, N, pH, PET, MAP, LC, AI, MAT, ELE, CEC, clay, silt | 12 |

Furthermore, we adopted a recently proposed variable selection method, namely forward recursive feature selection (FRFS) to reduce the number of predictor variables while not losing model performance (Xiao et al., 2022; Zhang et al., 2023). FRFS employs a forward selection strategy, involving the following sequential steps: (1) initially, a RF model is fitted using all the $n$ predictors, and their variable importance is calculated; (2) the most important predictor (only one) is selected to create an initial model, and its performance is assessed using 10-fold cross-validation with a single predictor in the pool; (3) subsequently, a series of models are constructed using two predictors, where the first predictor is chosen from the pool, and the second predictor is selected from the remaining predictors. The model performances are evaluated, and the model with the best performance is recorded; (4) the pool of predictors is then updated based on the predictors from the best-performing model in the previous step; (5) The process is iteratively repeated, progressively increasing the number of predictors from 3 to $n$. Ultimately, the predictors used in the model with the best performance are selected to form the final predictive model, as detailed in the work of Xiao et al. (2022). The R script for implementing FRFS is accessible at https://doi.org/10.5281/zenodo.7141020. In this study, FRFS was applied to select the most relevant predictors constructing the predictive models (Table 3).

For clarity, in global modelling, we refer to the RF model using the full variables as global-$RF_{FULL}$, and the combination of RF with FRFS as global-$RF_{FRFS}$.

**2.5 Local ML-PTFs**

The development of local ML-PTFs consists of four steps: (1) use the Mahalanobis distance to calculate the distances of predictor variables between each sample to be predicted and all the samples in the database; (2) select $k$ nearest neighbour samples to fit a RF model for each unknown sample; (3) predict the $BD_{fine}$ for each unknown sample using relevant RF models. Since the number of nearest neighbour samples ($k$) is an important parameter in the local model, we evaluated its effect on the model performance by testing $k$ from 20 to 700 (20, 40, 60, 80, 100, 150, 200, 250, 300, 350, 400, 450, 500, 550, 600, 650, 700).

For clarity, we refer to the local modelling using the full variables as local-$RF_{FULL}$, and for the combined use of RF and variables selected by global-$RF_{FRFS}$, we refer it as local-$RF_{FRFS}$.

**2.6 Model evaluation**

Due to the large sample size, single random split is stable compared to k-fold cross-validation or repeated random split (Chen et al., 2021). Therefore, we used randomly split (80% for calibration and 20% for validation) to assess the model performance of earlier published PTFs and ML-PTFs. It is important to note that the same validation set was used to evaluate earlier published PTFs and ML-PTFs. The root mean square error (RMSE), determination coefficient ($R^2$) and relative error (RE) were used as performance indicators on the validation set (Chen et al., 2022). These indices are defined as following Eq. (1), (2) and (3):

$$RMSE = \sqrt{\frac{1}{n}\sum_{i=1}^{n}(O_i - P_i)^2} \tag{1}$$

$$R^2 = 1 - \frac{\sum_i^n(O_i - P_i)^2}{\sum_i^n(O_i - \bar{O})^2} \tag{2}$$

$$RE = \frac{1}{n}\sum_{i=1}^{n}\frac{|O_i - P_i|}{O_i} \times 100\% \tag{3}$$

where $n$ represents the number of observations, $O_i$ and $P_i$ are the observed and predicted $BD_{fine}$ for observation $i$, and $\bar{O}$ is the mean of the observed $BD_{fine}$. A good model has RMSE and RE close to 0, and also higher $R^2$ close to 1.

**2.7 The build-up of extended $BD_{fine}$ and SOC stock datasets for topsoil in Europe**

Since only part of LUCAS 2015/2018 had soil particle fractions and $CaCO_3$, we used the unique samples ID to link the missing soil particle fractions and $CaCO_3$ using LUCAS Soil 2009 for the same sampling sites. This operation is reasonable since soil particle fractions and $CaCO_3$ will not have a notable change within a decade. The SOC stock (kg m$^{-2}$) at a depth of 0-20 cm for LUCAS Soil 2018 was calculated by the SOC content (g kg$^{-1}$), $BD_{fine}$ (g cm$^{-3}$), $CF_{volumefraction}$ (%/100), and depth (20 cm) as Eq. (4) (Poeplau et al., 2017).

$$SOC\ stock = SOC \times BD_{fine} \times Depth \times (1 - CF_{volumefraction})/100 \tag{4}$$

**3 Results**

**3.1 Statistics of $BD_{fine}$ and its correlation with predictor variables**

Fig. 2 illustrates the histogram of $BD_{fine}$ values and their distribution in a ternary soil texture triangle. The dataset consists of 5,163 topsoil samples with $BD_{fine}$ ranging from 0.20 to 1.89 g cm$^{-3}$. The topsoil sample with the lowest $BD_{fine}$ (0.20 g cm$^{-3}$) was collected from Pine dominated mixed woodland with a SOC content greater than 137 g kg$^{-1}$. In contrast, the topsoil sample with the highest $BD_{fine}$ (1.89 g cm$^{-3}$) was sampled from a sandy soil (sand and clay of 65% and 11%, SOC content of 31.9 g kg$^{-1}$) in cropland (common wheat). Approximately half of the topsoil samples exhibited $BD_{fine}$ between 0.8 and 1.4 g cm$^{-3}$, while less than 10% of the topsoil samples had $BD_{fine}$ exceeding 1.4 g cm$^{-3}$. As shown in the soil texture triangle, the selected topsoil samples covered a wide range of soil texture classes.

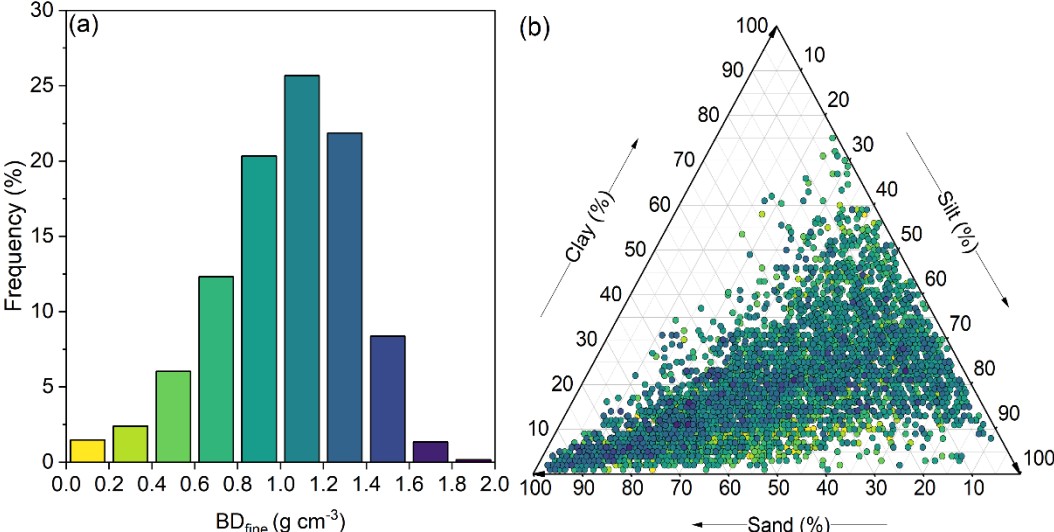

Figure 2 Histogram of $BD_{fine}$ (a) and USDA soil texture triangle (b). The point colors shown in the texture triangle correspond to the colors present in the left histogram. The percentage of each bin is indicated over the bin in the histogram.

Fig. 3 depicts the correlation matrix between $BD_{fine}$ and 15 predictor variables. $BD_{fine}$ exhibited positive correlations with pH and MAT, with correlation coefficients (r) greater than 0.25. On the other hand, $BD_{fine}$ showed notably high negative correlations with most of the other predictors. The most influential negative predictor was SOC content (r=-0.62), followed by N (r=-0.56), and $CaCO_3$ (r=-0.33). Note that $BD_{fine}$ under various LC classes exhibited significant differences with mean $BD_{fine}$ of 1.16, 1.00, 0.78, and 1.02 g cm$^{-3}$ for cropland, grassland, woodland and others, respectively.

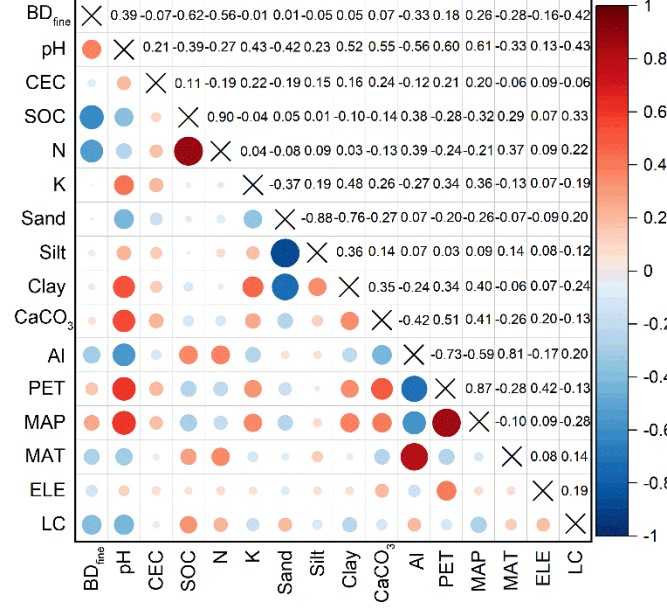

Figure 3 Correlation plot among $BD_{fine}$ and predictors. The sizes of the circle represent the magnitudes of the correlation, and light and dark colors represents negative and positive correlation respectively. The abbreviations are detailed below: $BD_{fine}$,
bulk density of fine earth; CEC, cation exchange capacity; SOC, soil organic carbon content; AI, aridity index; PET, potential evapotranspiration; MAP, mean annual precipitation; MAT, mean annual temperature; ELE, elevation; LC, land cover.

## 3.2 Selection of predictor variables

Table 3 presents the predictor variables utilized in the RF model for predicting $BD_{fine}$. In the global-$RF_{FULL}$ model, 15 predictor variables were included, namely clay content, silt content, sand content, pH, SOC content, $CaCO_3$, N, K, CEC, AI, PET, ELE,
MAP, MAT, and LC. On the other hand, the global-$RF_{FRFS}$ identified a subset of 8 predictor variables by FRFS that were deemed most important for $BD_{fine}$ prediction. These selected predictor variables, ranked in descending order of importance, were SOC content, N, pH, PET, MAP, LC, AI, and MAT.

## 3.3 Comparison of ML-PTFs and earlier published PTFs in $BD_{fine}$ prediction

In this study, we compared ML-PTFs with four earlier published PTFs in $BD_{fine}$ prediction (Fig. 4 and Fig. 5). The earlier
published PTFs had model performances with RMSE of 0.22 g cm$^{-3}$, $R^2$ of 0.40-0.45, and RE of 19.11-20.75%. The global-RF models had higher model performance with RMSE of 0.19 g cm$^{-3}$, $R^2$ between 0.57 and 0.58, and RE of 16.53-16.74% for global-$RF_{FULL}$ and global-$RF_{FRFS}$ respectively, whereas the later performed slightly better (see $R^2$ values in Fig. 4). As for local models, it was clear that the model performance showed an increasing trend when the number of neighbour samples increased and some fluctuations were observed after the model performance reached a plateau. The number of neighbour samples were
optimized at 350 and 400 for local-$RF_{FRFS}$ and local-$RF_{FULL}$, respectively. Compared to global modelling, the best local-$RF_{FRFS}$ and local-$RF_{FULL}$ performed slightly better with $R^2$ of 0.59-0.57 and RE of 16.28-16.47%.

The summary of RE variations under different $BD_{fine}$ levels and land covers using best earlier published PTF (PTF-4) and ML-PTF (local-$RF_{FRFS}$) is shown in Fig. 6. The results indicated that local-$RF_{FRFS}$ (RE of 29%) performed much better than PTF-4 (RE of 37%) for the topsoil with low $BD_{fine}$ (<0.8 g cm$^{-3}$). The improvement of RE for other BD levels was rather limited
($\Delta$RE of 1-3%). The highest RE (30-57% for PTF-4, 25-50% for local-$RF_{FRFS}$) was found for topsoil with low $BD_{fine}$ for the whole validation set and each land cover. Across land covers, the RE generally decreased greatly (15-24% for PTF-4, 14-20% for local-$RF_{FRFS}$) for topsoil with low-median $BD_{fine}$ (0.8-1 g cm$^{-3}$), and then to its lowest (7-9% for both PTF-4 and local-$RF_{FRFS}$) for topsoil with median-high $BD_{fine}$ (1-1.2 g cm$^{-3}$). A slight increase of RE (14-16% for PTF-4, 11-17% for local-$RF_{FRFS}$) was observed for topsoil with high $BD_{fine}$ (>1.2 g cm$^{-3}$) for all the land covers. Among different land covers, the
cropland had the greatest RE for topsoil with low and low-median $BD_{fine}$, followed by others, woodland and grassland. For topsoil with median-high and high $BD_{fine}$, a similar RE was found for all the land covers. Overall, the RE for both PTF-4 and local-$RF_{FRFS}$ showed the worse performances for low $BD_{fine}$, but the results were always better for local-$RF_{FRFS}$, except for woodlands with $BD_{fine}$>1 where the RE was slightly better for PTF-4.

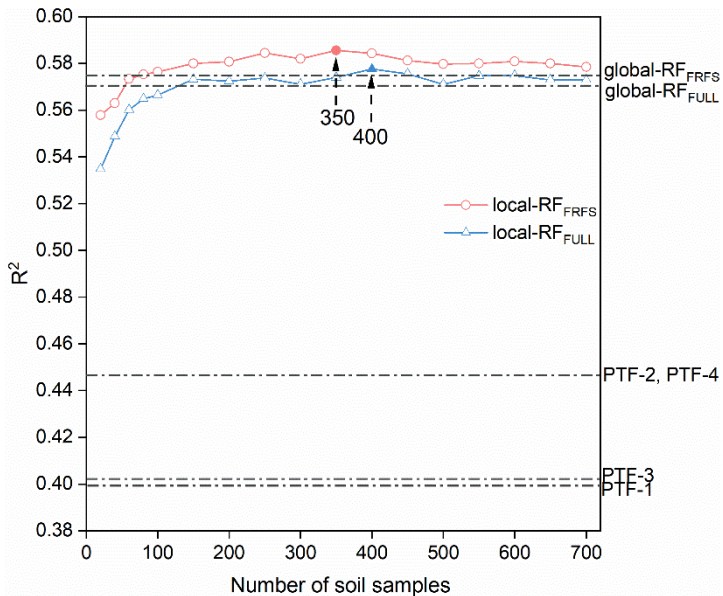

**Figure 4** Model performance indicator ($R^2$) of earlier published PTFs and ML-PTFs in $BD_{fine}$ prediction. The performances of local RF models (local-$RF_{FULL}$ and local-$RF_{FRFS}$) change with the number of soil samples used for local modelling.

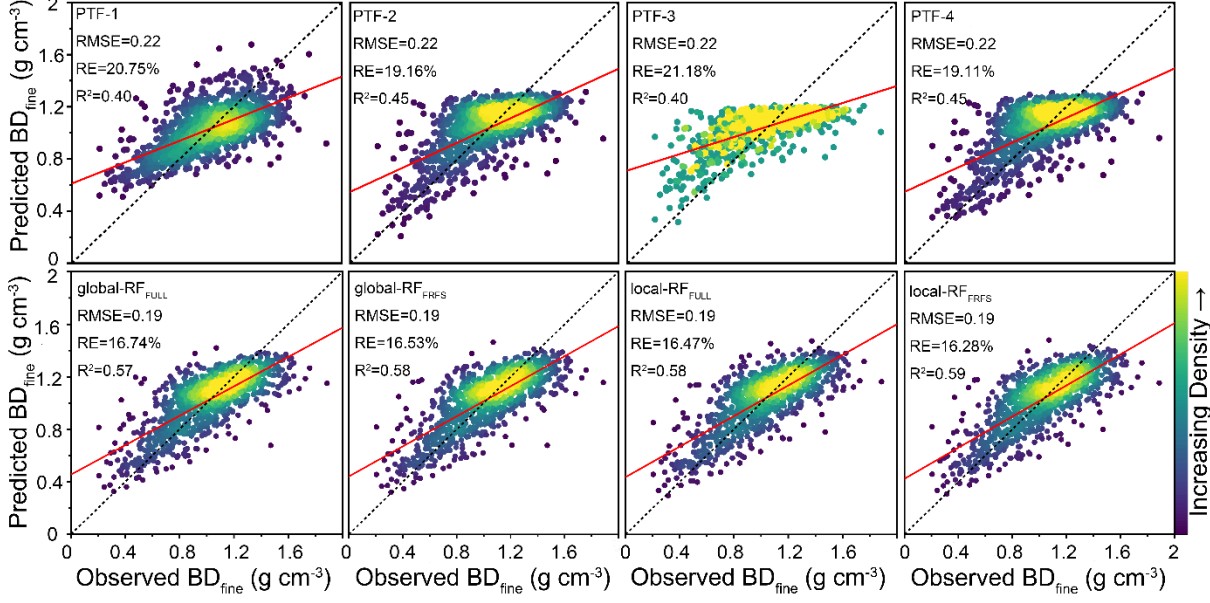

**Figure 5** Scatter plots of $BD_{fine}$ predictions using earlier published PTFs and ML-PTFs along with model performance indicators (RMSE, $R^2$ and RE). The lighter color means higher sample density. Please note that the best models are selected for local-$RF_{FULL}$ and local-$RF_{FRFS}$.

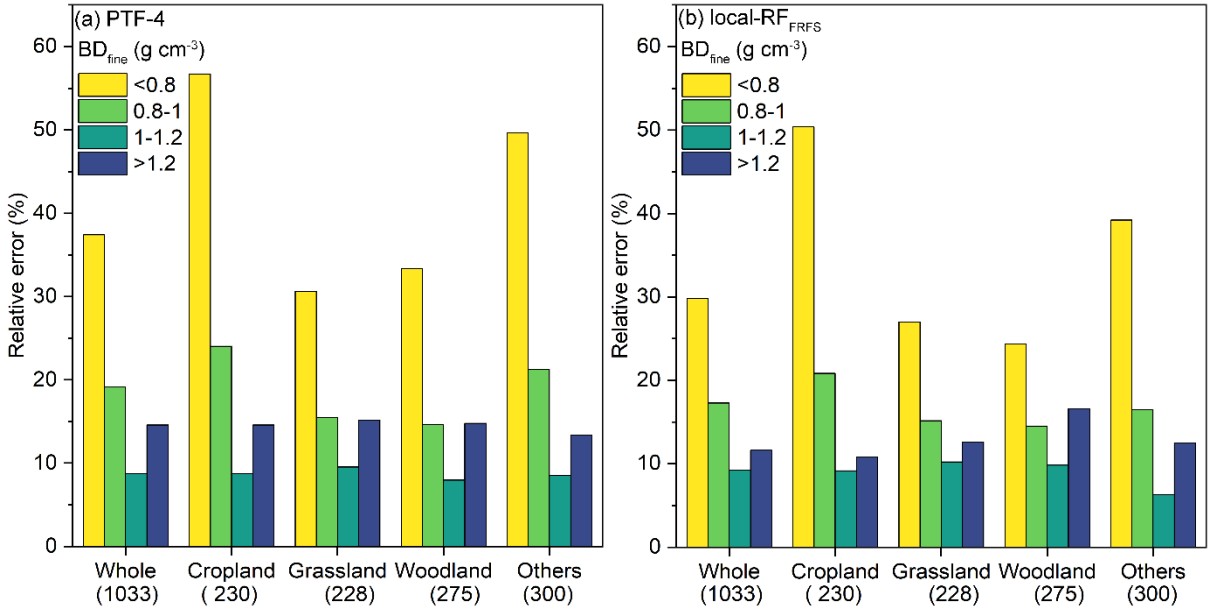

**Figure 6** The variations of RE related to $BD_{fine}$ ranges of values (<0.8, 0.8-1, 1-1.2 and >1.2 g cm$^{-3}$) and land covers using PTF-4 (a) and local-RF$_{FRFS}$ (b). The number under the land cover is the corresponding topsoil sample size.

### 3.4 Comparison of ML-PTFs and earlier published PTFs in SOC stock calculation

We investigated how using $BD_{fine}$ estimated by PTFs impacted the accuracy of SOC stock calculation (Fig. 7). We found that SOC stock calculation using $BD_{fine}$ predictions from four earlier published PTFs resulted in a good performance with RMSE of 1.39-1.89 kg m$^{-2}$, $R^2$ of 0.70-0.84, and RE of 17.57-19.46%, respectively. Meanwhile, the performance indicators of SOC stock calculation using $BD_{fine}$ prediction (RMSE of 1.32-1.36 kg m$^{-2}$, $R^2$ of 0.84-0.85, RE of 15.01-15.41%) exhibited always slightly better performances than the earlier published PTFs. However, the performances of the best earlier published PTF (PTF-4) were rather similar to those of the local-RF$_{FRFS}$. Overall, the performances of the local-RF$_{FRFS}$ were the best.

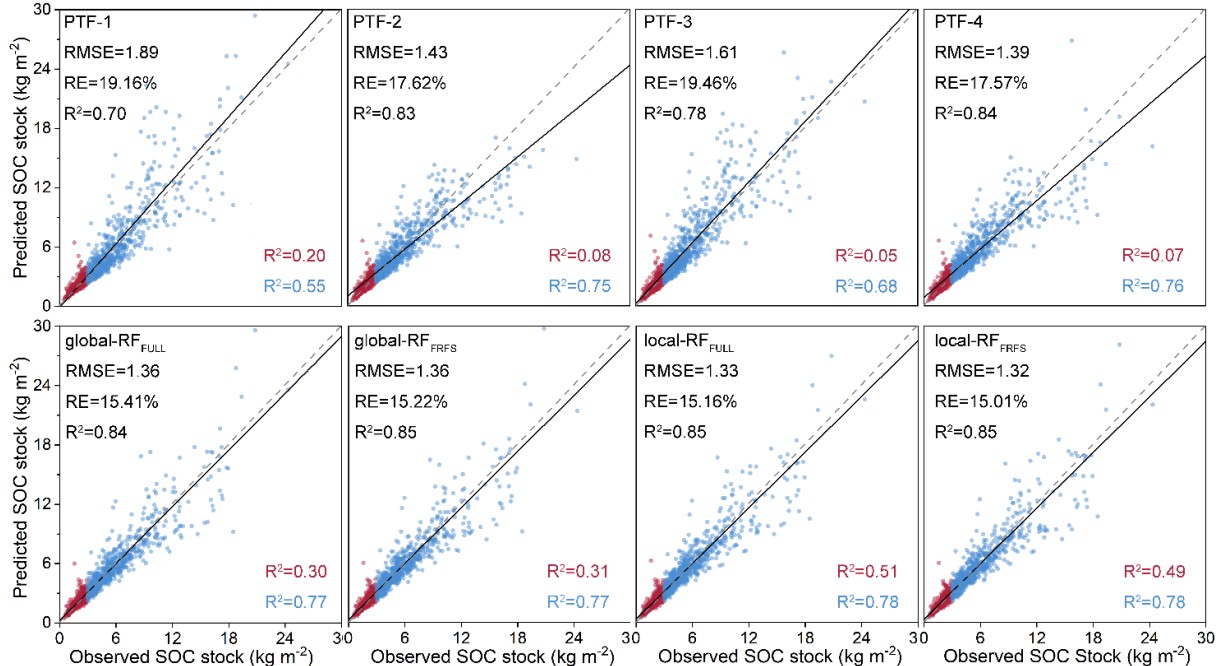

**Figure 7** Scatter plots of SOC stock predictions by earlier published PTFs and ML-PTFs along with model performance indicators (RMSE, $R^2$ and RE). The red points represent topsoil samples with SOC stock<3 kg m$^{-2}$ while the blue points represent topsoil samples with SOC stock $\geq$ 3 kg m$^{-2}$. Note that observed SOC stock is computed using SOC content, $CF_{volumefraction}$, $BD_{fine}$ observations, and while predicted SOC stock is computed using SOC content observations, $BD_{fine}$ predictions and $CF_{volumefraction}$ transformed from $CF_{massfraction}$ using $BD_{fine}$ predictions suggested by Pacini et al. (2023).

### 3.5 Summary of the extended European topsoil $BD_{fine}$ and SOC stock database

To enlarge the topsoil $BD_{fine}$ and SOC stock database (0-20 cm) for the Europe, we refitted the best ML-PTF (local-RF$_{FRFS}$) and the best earlier published PTF (PTF-4) using all the 5,163 topsoil samples to predict topsoil samples without $BD_{fine}$ and then calculated SOC stock, which resulted in 15,389 and 18,945 topsoil samples predictions for the extended database respectively (less topsoil samples had all the required variables for the use of local-RF$_{FRFS}$). As shown in Fig. 8, these extended topsoil $BD_{fine}$ and SOC stock databases are more regularly distributed across EU and UK compared to the points in Fig. 1. In EU and UK, $BD_{fine}$ in topsoil was primarily distributed within 1.0-1.2 g cm$^{-3}$ (46-47%) while the SOC stock in topsoil was mainly comprised between 2 and 4 kg m$^{-2}$.

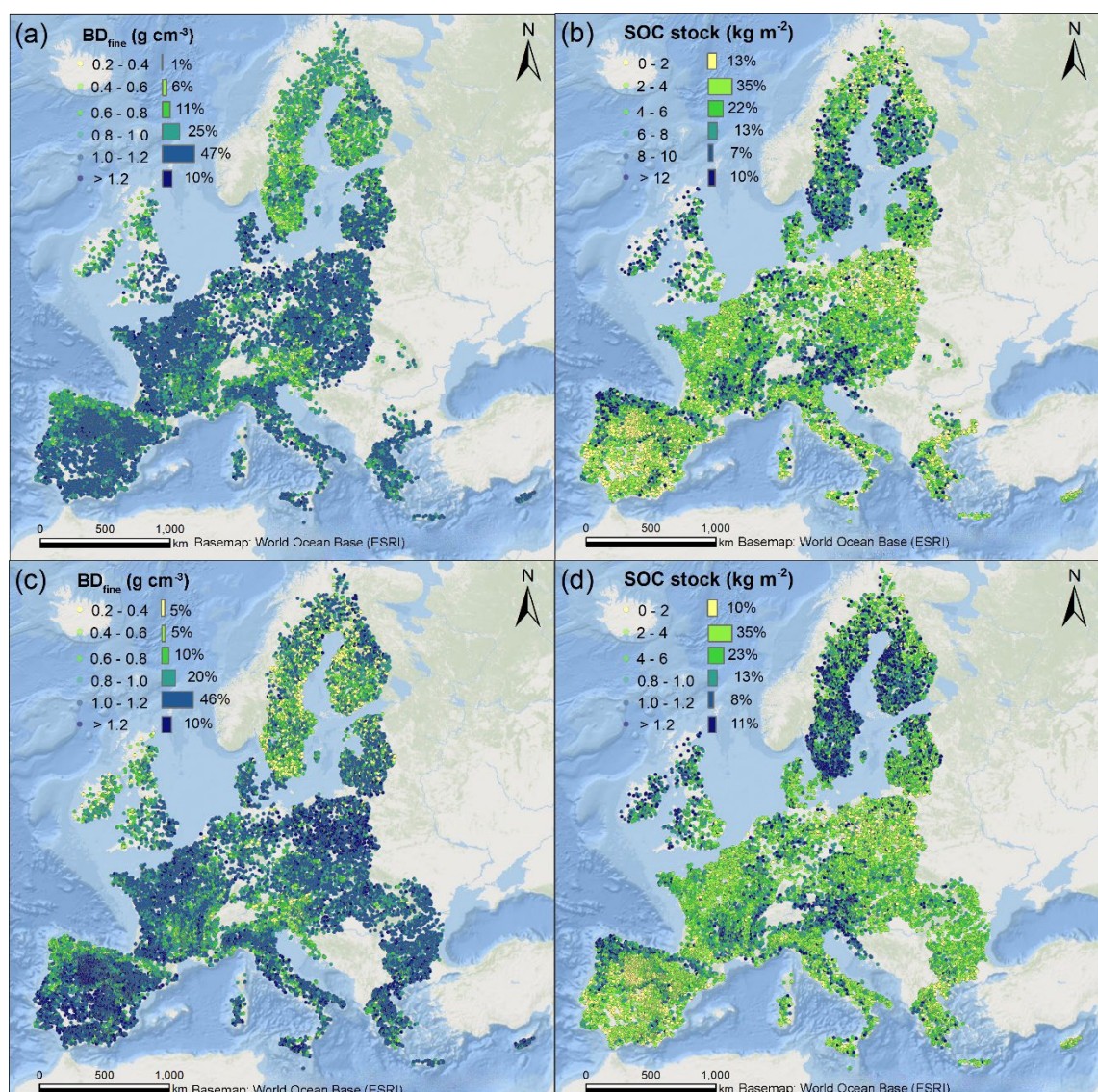

**Figure 8** Spatial distributions of 15,389 topsoil samples with $BD_{fine}$ (a) and SOC stock (b) from LUCAS 2018 Soil using local-$RF_{FRFS}$, and 18,945 topsoil samples with $BD_{fine}$ (c) and SOC stock (d) from LUCAS 2018 Soil using PTF-4.

As shown in Fig. 9, in the database created by local-$RF_{FRFS}$ (15,389 topsoil samples), the topsoil samples under cropland had the highest median $BD_{fine}$ of 1.11 g cm$^{-3}$, while woodland exhibited the lowest median $BD_{fine}$ at 0.84 g cm$^{-3}$. Conversely, woodland had the highest median SOC stock at 6.21 kg m$^{-2}$, while cropland showed the lowest median SOC stock at 3.06 kg m$^{-2}$. As for the database built on PTF-4 (18,945 topsoil samples), cropland also had the highest median $BD_{fine}$ at 1.14 g cm$^{-3}$

while woodland exhibited the lowest median $BD_{fine}$ at 0.86 g cm$^{-3}$. In contrast, the SOC stock under woodland presented the highest median SOC stock at 6.96 kg m$^{-2}$ while cropland had the lowest median SOC stock at 3.17 kg m$^{-2}$.

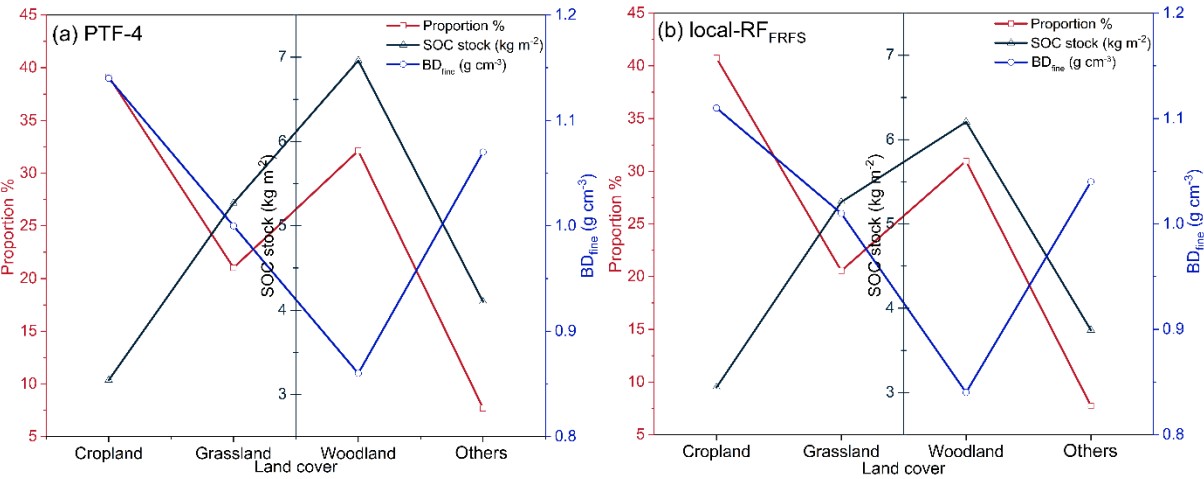

Figure 9 Variations of topsoil $BD_{fine}$ and SOC stock under different land covers using PTF-4 (a) and local-RF$_{FRFS}$ (b).

## 4 Discussion

### 4.1 The superiority of ML-PTFs in BD$_{fine}$ prediction

In this study, using the LUCAS Soil and 15 predictor variables, we compared the model performance of four earlier published PTFs and four ML-PTFs for $BD_{fine}$ in topsoil (0-20 cm). Four earlier published PTFs showed a moderate model performance with $R^2$ of 0.40-0.45, which is close to a recent developed Hollis-type PTF ($R^2$ of 0.41, Hollis et al., 2012) that refitted by LUCAS Soil 2018 data (De Rosa et al., 2023). Our results underscored the efficacy of ML-PTFs in successfully predicting $BD_{fine}$ at a continental scale, yielding a substantial $R^2$ ranging from 0.57 to 0.59. It indicates that when adding more relevant predictor variables (e.g., N, pH, PET, MAP) in the topsoil database, ML-PTFs is a better choice for improving $BD_{fine}$ prediction than earlier published PTFs based on algebraic equations. Otherwise, earlier published PTFs are still the best choice to impute the missing data due to their simplicity (Van Looy et al., 2017).

In addition to global PTFs that use all the soil samples, we introduced the local modelling strategy in PTFs which searched similar samples first and then built the relevant PTF for each unknown sample dynamically. Generally, the model performance of local PTFs (local-RF$_{FULL}$ and local-RF$_{FRFS}$) for $BD_{fine}$ prediction continuously improved with the increasing number of neighbour samples, and then it reached a plateau when number of neighbour samples reached approximately 350 to 400 (Fig. 4). Compared to the global PTFs (4,500 soil samples), the size of local PTFs were much smaller (350-400 soil samples) with slightly better model performance. Therefore, the comparison between global PTFs and local PTFs performances shows that local PTFs can improve the efficiency for imputing missing data using a large soil database (Padarian et al., 2019; Sanderman et al., 2020).

Comparing with the earlier published PTFs that were refitted using our data, the local-FR$_{FRFS}$ model substantially improved model performance in BD$_{fine}$ prediction ($\Delta R^2$ of 0.14-0.19). Our results suggest that ML-PTFs performed much better than earlier published PTFs for BD$_{fine}$ prediction. This resulted from the fact that most of ML models are able to handle non-linear and complex relationships between the predictor variables and the response variable so as to improve predictions compared to those of earlier published PTFs (Katuwal et al., 2020; Palladino et al., 2022). Meanwhile, the earlier published PTFs typically rely solely on SOC or SOM content for BD$_{fine}$ prediction. This approach maintains model simplicity but overlooks readily available predictor variables such as particle size fractions, MAT and MAP, which are also pertinent to BD$_{fine}$ prediction (Abdelbaki, 2018). Despite of the high diversity in landscapes and climates at a continental scale, the proposed local-FR$_{FRFS}$ model demonstrated similar or even superior performance compared to the ML-PTFs conducted at regional and national scales (Table 1).

Looking into the RE for topsoil under different BD$_{fine}$ levels (Fig. 6), it is clear that the fitted best PTFs (PTF-4 and local-RF$_{FRFS}$) had the highest REs for topsoil with low BD$_{fine}$ (<0.8 g cm$^{-3}$) despite that local-RF$_{FRFS}$ performed better. This partly results from the low BD$_{fine}$ to calculate the RE, because BD$_{fine}$ value is used as the reference 100% value in RE calculation. This is also likely due to the general trend of broad-scale predictions to smooth the variability and to overestimate the low values and to underestimate the high values whatever the predicted variable is (e.g., Tifafi et al., 2018; Lemercier et al. 2022; Richer-de-Forges et al., 2023). Most important, many low BD$_{fine}$ observations are probably linked to large voids resulting in a large porosity, especially under disturbed topsoil. This explains why cropland topsoil exhibited such a large RE, likely due to the effect of soil tillage which cannot be predicted by our predictor variables. This can also explain the decreasing trend of RE with the increase of BD$_{fine}$ up to 1.2 g cm$^{-3}$ whereas for the topsoil with high BD (>1.2 g cm$^{-3}$), both local-RF$_{FRFS}$ and PTF-4 showed a slight increase in RE. Overall, the RE might appear a bit deceiving if we compare them to the accuracy that one may wish for monitoring changes in BD$_{fine}$ for example as an indicator of compaction. We must state that this is clearly out of the scope of this study, which is to provide a wide database that can be used for broad-scale modelling.

## 4.2 Performance of FRFS and variable importance in BD$_{fine}$ prediction by ML-PTFs

We reduced the number of predictor variables in RF model from 15 to 8 using the FRFS algorithm, and the model performance of global-RF$_{FRFS}$ for BD$_{fine}$ using FRFS selected variables was higher than global-RF$_{FULL}$ using full variables (Table 3). Though the local-RF$_{FRFS}$ (R$^2$ of 0.59) only had marginal superiority over the local-RF$_{FULL}$ model (R$^2$ of 0.58), it facilitated the reduction of variables, consequently enhancing prediction efficiency (Fig. 4 and 5). This outcome validates the capacity of FRFS to simplify the model complexity while concurrently enhancing predictive accuracy (Xiao et al., 2022; Liu et al., 2023; Zhang et al., 2023; Hu et al., 2024). Being a useful tool for gap-filling the missing data, an ideal PTF requires both high parsimony and good fit. If the developed PTF needs too many predictors variables, its practical applicability would be limited, as much fewer soil samples have all the required predictors variables.

## 4.3 The build-up of extended $BD_{fine}$ and SOC stock datasets in Europe

We used the $BD_{fine}$ predictions from eight PTFs together with $CF_{volumefraction}$ to calculate the SOC stock. The result showed that the model performances of SOC stock ($R^2$ of 0.70-0.85) were much higher than those of $BD_{fine}$ ($R^2$ of 0.40-0.59) (Fig. 5 and Fig. 7). It can be explained by the interdependence between $BD_{fine}$ and SOC content. For instance, a soil sample with a high SOC content commonly has a large pore space due to the large amount of organic matter, leading to a low $BD_{fine}$ (Perie and Ouimet, 2008; Chen et al., 2018). As shown in Fig.7, high SOC content and $BD_{fine}$ were always underestimated while the low SOC content and $BD_{fine}$ were overestimated. By multiplying these two negatively correlated variables, the predicted SOC stock could be closer to the observed SOC stock as the overestimation (underestimation) of $BD_{fine}$ can counterbalance the underestimation (overestimation) of SOC content, resulting in better model performance than $BD_{fine}$. It is interesting to note that the model performance of best earlier published PTFs (PTF-4, $R^2$ of 0.84) and ML-PTFs (local-RF$_{FRFS}$, $R^2$ of 0.85) was quite close in SOC stock prediction. This indicated that the improvement of $BD_{fine}$ prediction by ML-PTFs did not impact the accuracy of SOC stock prediction. Looking into the scatter plots shown in Fig. 5, we can observe that the ML-PTFs performed much better than earlier published PTFs for topsoil samples with high $BD_{fine}$ (and low SOC content) while limited difference was found for soil samples with low $BD_{fine}$ (and high SOC content). Compared to earlier published PTFs, ML-PTFs tended to predict SOC stock better for topsoil samples with low SOC stock ($<3$ kg m$^{-2}$) while similar model performance can be found in topsoil samples with high SOC stock ($\geq 3$ kg m$^{-2}$), which is evident in Fig. 5. As a result, the best earlier published PTF (PTF-4) performed quite similar to the best ML-PTF (local-RF$_{FRFS}$) when considering the topsoil samples with a wide range of SOC stock. This last result suggests that earlier published PTFs could be useful default tools to estimate $BD_{fine}$ which is subsequently used for SOC stock calculation. One of the advantages of these earlier published PTFs is their simplicity; another obvious advantage is that they require less training soil samples than ML-PTFs to be fitted and validated. Otherwise, if enough data is available, ML-PTFs are suggested for more accurate $BD_{fine}$ prediction, especially for regions with low SOC stock such as dry land regions in Spain and Italy (Maestre et al., 2021; De Rosa et al., 2023; Wang et al., 2023).

## 4.4 Limitations and perspectives

It is essential to acknowledge that our developed PTFs for $BD_{fine}$ prediction was constructed based on LUCAS Soil data (0-20 cm), confining its applicability to topsoil within the EU and UK (Orgiazzi et al., 2022, Panagos et al., 2022). However, the potential of their extrapolation capability to other regions or to deep soil ($>20$ cm) necessitates further evaluation. As more soil data become available from diverse regions as well as for deep soil (Lal, 2018; Tautges et al., 2019; Batjes et al., 2020; Yost et al., 2020; Palmtag et al., 2022; Armas et al., 2023), the proposed methodology can be further used to update the PTFs, thereby broadening their area of applicability (Chen et al., 2018; Meyer and Pebesma, 2021). In addition, when a depth-specific soil $BD_{fine}$ database is available, it will be important to develop depth-explicit ML-PTFs to account for the effects of climate and topography on $BD_{fine}$ at depths.

We acknowledge that our use of PTF-3 and PTF-4 is based on measured SOC contents and on a fixed Van Bemmelen factor (SOM=1.724×SOC, Sprengel, 1826; Van Bemmelen, 1890). One good reason to use this factor is that it enables a comparison with most of the studies predicting $BD_{fine}$ using SOC and other soil properties. One pitfall is that we know that the conversion factor from SOC to SOM is not constant (Pribyl, 2010). However, this conversion factor was only used for PTF-3 and PTF-4.

Considering the equations used, changing this conversion factor for PTF-4 has no consequence on the predicted $BD_{fine}$, neither on the model performance of the PTF for $BD_{fine}$ prediction. Changing it for PTF-3 will lead to lower performance. We have no clear indication to try to adapt the Van Bemmelen factor to the pedological context (neither the effect of SOC on $BD_{fine}$) when we used fixed regressions such as PTF-3 and PTF-4. One advantage of ML-PTFs and especially of local ML-PTFs is that they can take into account interactions between soil properties. Therefore, the importance of SOC likely varies depending

other local controlling factors such as clay content, climate or even the nature of the organic compounds, which could explain the strong effect of N. In other words, ML-PTFs were able to partially compensate for the effect of using a fixed conversion factor between SOC and SOM. It should be noted that the $BD_{fine}$ and $CF_{volumefraction}$ used in this study have been transformed from $BD_{sample}$ and $CF_{massfraction}$ by Pacini et al. (2023), which certainly introduced some uncertainty. However, for topsoil samples with CF close to 0, the uncertainty from data transformation is rather low. Since many cropland soils have CF close

to 0, and they are the most sensitive to threats, the proposed PTFs for $BD_{fine}$ prediction would be helpful.

Another possible source of error is linked to re-allocating some measured values from one LUCAS Soil sampling campaign to another one. Indeed, BD (whether $BD_{fine}$ of $BD_{sample}$) is highly variable in space and time, and coarse fragments and SOC are highly variable in space. The location of sampling may have slightly change between LUCAS Soil campaigns for various reasons and the instructions recommend a distance <100m (Fernández-Ugalde et al., 2017). This latter case has no reason to

385 induce a systematic bias. However, it increases the uncertainty (Munera-Echeverri et al., 2022). Finally, soils containing large amounts of large rocks are clearly excluded from the LUCAS Soil protocol, therefore one should keep this in mind not to extrapolate $BD_{fine}$ and SOC stocks predictions to rocky soils.

If ones want to use PTFs based $BD_{fine}$ prediction to detect SOC stock changes, the impact of the performance of PTFs on the accuracy of SOC stock calculation remains unclear since the equivalent soil mass approach also require $BD_{fine}$ as input

(Schrumpf et al., 2011; Wendt and Hauser, 2013). Therefore, this issue could be investigated in future studies. However, the most straightforward and unbiased way to measure SOC stocks by sampling remains the direct determination of the ratio fine-earth mass:sample volume by sieving and weighting the fine soil from a sample of know volume.

Most of the predictor variables that we used for ML-PTFs are prone to changes at different time scales. This is the case for all predictor variables derived from climate. Some soil predictor variables (e.g., SOC, pH) can change more-or-less rapidly under

395 the effect of practices, LC changes and global changes. Finally, LC can change a given time, though some effect of past LC may remain for a given time. Though strong perturbations may have an immediate effect on $BD_{fine}$, the time-scales at which most of these predictor variables influence or are just correlated to $BD_{fine}$ remain unclear. This opens the door to further questioning about the processes that govern the importance of these predictor variables on $BD_{fine}$. Indeed, ML tools can be

used as simple predictors at a given time, or as tools to raise attention to the possible effects of some controlling factors and their changes, and to the processes involved in these effects.

## 5 Data availability

All the soil data used in this article are available at the following data sources: (1) Land Use and Coverage Area Frame Survey Soil (LUCAS Soil) 2009 via https://esdac.jrc.ec.europa.eu/content/lucas-2009-topsoil-data (Panagos et al., 2022), (2) LUCAS Soil 2015 via https://esdac.jrc.ec.europa.eu/content/lucas2015-topsoil-data (Fernández-Ugalde et al., 2022), (3) LUCAS Soil 2018 via https://esdac.jrc.ec.europa.eu/content/lucas-2018-topsoil-data (Panagos et al., 2022), (4) the European topsoil $BD_{fine}$ and SOC stock dataset (0-20 cm) in this paper is available at https://zenodo.org/records/10211884 (Chen et al., 2023a).

## 6 Conclusions

Using the largest extendable soil dataset for Europe, we have developed ML-PTFs for predicting $BD_{fine}$ at 0-20 cm across the EU and UK. In comparison with four earlier published PTFs, the best ML-PTF, namely local-$RF_{FRFS}$, exhibited superior performance for $BD_{fine}$ prediction with percentage increase in $R^2$ at 31.1-47.5%, percentage decrease in RMSE and RE at 13.6% and 14.8-23.1%, respectively. When the predicted $BD_{fine}$ was subsequently used for SOC stock calculation, we found that the best earlier published PTF preformed quite similar to the best ML-PTF, indicating the fact that earlier published PTFs would be useful for $BD_{fine}$ prediction when targeting in SOC stock calculation. However, for regions with low SOC stock (<3 kg m$^{-2}$), ML-PTFs are still recommended due to its high accuracy in SOC stock calculation. Finally, we established two comprehensive pan-European topsoil $BD_{fine}$ and SOC stock databases (0-20 cm) including 15,389 and 18,945 soil samples in LUCAS Soil 2018 using the best ML-PTF (local-$RF_{FRFS}$) and earlier published PTF (PTF-4), respectively. Our study proposed a potential model to improve the performance of $BD_{fine}$ prediction, and the resultant topsoil $BD_{fine}$ and SOC stock datasets at 0-20 cm across the EU and UK enable more precise soil hydrological and biological modelling at a continental scale.

## 7 Author contributions

SC, ZC and XZ compiled the data. SC and ZC performed the analysis and drafted the manuscript. XZ, ZL, CS, DA, ACRdF and ZS validated the results and revised the manuscript. SC acquired of the financial support. ZS supervised this work.

## 8 Competing interests

The contact author has declared that none of the authors has any competing interests.

## 9 Disclaimer

## 10 Financial support

This study is funded by the National Natural Science Foundation of China (No. 42201054).

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
