# Peer review of "European soil bulk density and organic carbon stock database using machine learning based pedotransfer function"

_Earth System Science Data, 2023_

## Author Comment (AC1)

**Responses to Reviewer 1**

Please note that your comments are provided in green text and our responses are marked in blue text. Our major modifications in the revised manuscript are marked as red text.

The manuscript from Chen et al. produced the European soil bulk density and organic carbon stock database (>15000 soil samples) using the recently released BDfine and CFvolumefraction data (around 6000 soil samples) from LUCAS 2018. Authors evaluated the model performance for BD using traditional pedotransfer functions (PTFs) and four proposed machine learning (ML) based PTFs, and found that ML based PTFs ($R^2$ of 0.56-0.57) greatly improved the accuracy for BD prediction, and this is also much higher than previous PTFs for Europe using Hollis-type PTF ($R^2$ of 0.41). For the first time, authors produced the European soil organic carbon stock data of topsoil (0-20 cm) for the year of 2018 and evaluated the impact of BD accuracy on the accuracy of soil organic carbon stock data. The produced data and relevant evaluation are of significant importance for informing more precise soil hydrological and biological modelling, so as to support Soil health by 2050 proposed by the European Commission. This manuscript is generally well-written with clear objectives and solid methodology, and therefore I suggest that it can be accepted for publication after minor revision.

Response: Many thanks for your high recognition on our work. We have carefully revised the manuscript based on your comments and suggestions. Please find our response to your concerns one by one below.

Lines 100-101: Several symbols for the units should be superscripts, such as g cm-3, g kg-1. Please correct them throughout the manuscript.

Response: Thanks for pointing out the typos for the units. We have carefully

checked the whole manuscript and corrected all the relevant issues.

Table 2: I think two digits would be enough for the R2 reported here, which is in-line with your previous summary in Table 1. It is also not clear whether the data used to evaluate these traditional PTFs are the same to machine learning PTFs? If not, the results would be not comparable. Please make it clear.

Response: Thank you for this nice suggestion. We have kept two digits for $R^2$ in Table 2 for the consistency in the revised manuscript. The earlier published PTFs and machine learning PTFs (ML-PTFs) have been validated on the same data to make them comparable. We have added relevant descriptions to make it clear in Lines 184-185: "It is important to note that the same validation set was used to evaluate earlier published T-PTFs and ML-PTFs.".

Line 140: Please specify the k here. 5-fold cross-validation? 10-fold cross-validation?

Response: Here 10-fold cross-validation have been used. We have specified it in Line 161.

Figure 6: What do the colours mean here? More details should be provided in the figure captions.

Response: We appreciate your helpful suggestion. The red points represent topsoil samples with SOC stock<3 kg m$^{-2}$ while the blue points represent topsoil samples with SOC stock≥3 kg m$^{-2}$. This information has been added in the caption of Figure 7 (previous Figure 6, in Lines 265-266).

---

## Author Comment (AC2)

**Responses to Reviewer 2**

Please note that your comments are provided in green text and our responses are marked in blue text. Our major modifications in the revised manuscript are marked as red text.

The authors presented a new model to estimate soil bulk density and evaluate the performance using the LUCAS database. Overall, the manuscript was well written. The improved model performance will significantly help the research community to derive more reliable soil carbon stock products. I have some minor comments below:

Response: We would like to thank your positive feedbacks on our manuscript. We have replied your detailed comments one by one below. Hope you are satisfied with our revision.

1. In the two existing bulk density models, the input data is SOM. Please explain how you estimate this from the LUCAS data where the listed variables only have SOC. If you used a conversion factor to estimate SOM from SOC, can you please run some simple uncertainty analysis for the four existing models? For example, if you replace the SOM input in models #3 and #4 with SOC, you will get two new models that only use SOC as inputs. Then, a simple inter-model comparison can be made by plotting the estimated bulk density from four models vs. measured bulk density using a range of SOC inputs. However, if you used independent sources of SOM for models #3 and #4, some of the uncertainty may be attributed to analytical errors related to SOM and SOC measurements.

Response: Thank you for this remark. We fully admit that the conversion factor from SOC to SOM is a matter of discussion (Pribyl, 2010). However, we think that we should clarify the point about changing SOM by SOC in PTF-3 and PTF-

4. We used SOC to estimate SOM by multiplying SOC by the conventional coefficient of 1.724. We fully acknowledge that this coefficient is not universal (Pribyl, 2010). However, it remains the most used and we have no real indication that can tell us how we could modify it in a clever way. Therefore, the most important point is to ensure that the same conversion coefficients are used throughout the model comparisons. We go now more in depth to the points about PTF-3 and PTF-4.

**PTF-3:** The first use of PTF-3 is not from Sun et al. (2020), neither from Mann (1986) but from Adams (1973). In their study, Sun et al. (2020) indicated that they "converted SOC to SOM by dividing a coefficient of 0.58" (Mann, 1986) when the SOM is not given in the literature". This conversion coefficient has been used for nearly 2 centuries (Sprengel, 1826, cited by Pribyl, 2010). It corresponds to SOM=1.724×SOC and is often called the "van Bemmelen factor" (Van Bemmelen, 1890). In a critical review, Pribyl (2010) stated that this conversion factor could range from 1.4 to about 2.5, and suggested that 1.24 might be on average to low and that 1.9 or 2.0 might be preferable. All the studies involved in the meta-analysis from Sun et al. (2020) used published SOC or SOM data which are from cultivated soils (either conventional or conservation tillage). Though it is not clearly specified in the paper from Sun et al. (2020), neither in its supplementary materials, it is likely that most of the original data used by Sun et al. (2020) came from SOC measurements, which are most of the time used to derive SOM (and not the reverse), especially in cultivated soils. In other words, some papers from the meta-analysis of Sun et al. (2020) used SOM for reporting, though they actually used SOC to estimate SOM by the coefficient of 1.724. When they had only SOC, Sun et al. (2020) used conversion to SOM too, because they used the original PTF from Adams (1973), and refitted it using their data. The Adams' equation refers to SOM because its aim is to consider the relative % of organic and mineral compounds in soil as if they were independent (except for the fact that they sum to 100%).

It is the reason why, in the Adams' ratio, 100% is used as numerator and SOM% and (100-SOM%) are used in the denominator. We have good reasons think that this Adam's equation used in PTF-3 is conceptually wrong because it does not consider interactions between organic and mineral compounds, but it is the basic hypothesis from Adams. Note that Adams constructed this equation in podzolic soils in which most SOM is particular and not complexed with clay minerals, especially in topsoil horizons (Jolivet et al., 1998). Knowing this hypothesis and the Adams' formula, using SOC% to replace SOM% in the PTF-3, using the 100% as a reference for the whole soil does not make sense from a conceptual and physical point of view, because the total does no more sum up to 100%. Doing this would make the implicit hypothesis that SOM=SOC and that the Van Bemmelen factor is equal to 1. To check it, we replaced SOM by SOC as you suggested and came to a strong decrease of $R^2$ from 0.41 to 0.31.

**PTF-4:** On the contrary, the PTF-4 does not make the same hypothesis based on (mineral compounds + organic compounds) = 100%. It just modulates a constant value of BD using an exponential decreasing effect SOM. Tao et al. (2023) used SOC content. To convert them into SOC stocks, they "used a pedo-transfer function to estimate the bulk density.

BD = $\alpha$ + $\beta$ × exp(−$\gamma$ × OM)

where OM is organic matter, calculated as SOC × 1.724".

Therefore, changing SOM by SOC does not change the fitting. Mathematically, using the equations

$$BD = a + b \times \exp(-c \times \%SOM)$$

or

$$BD = a + b \times \exp(-c \times \%SOC)$$

will result exactly in the same fitting, but with a change of the coefficient *c* which will be multiplied by 1.724 (or another value of the Van Bemmelen factor if we have a good reason to use this value). Therefore, in this case the $R^2$ remains the same and the predicted values of BD do not change, but, of course, the coefficient *c* of the equation changes. Thus, we have added relevant parts in

the discussion in Lines 366-377: "We acknowledge that our uses of PTF-3 and PTF-4 are based on measured SOC contents and on a fixed Van Bemmelen factor (SOM=1.724×SOC, Sprengel, 1826; Van Bemmelen, 1890) to convert them into SOM. One good reason to use this factor is that it enables a comparison with most of the studies predicting BD using SOC and other soil properties. One pitfall is that we know that the conversion factor from SOC to SOM is not constant (Pribyl, 2010). First, we recall that we used this conversion factor for PTF-3 and PTF-4 only. Besides, considering the equations used, changing this conversion factor for PTF-4 has no consequence on the predicted absolute values of BD, neither on the goodness of the fit of the PTFs predicting BD. On the contrary, changing the conversion factor for PTF-3 will have consequences. We have no clear indication to try to adapt the Van Bemmelen factor to the pedological context (neither the effect of SOC on BD) when we use fixed regressions such as in PTF-1, PTF-2, PTF-3, and PTF-4. One advantage of ML-PTFs and especially of local ML-PTFs is that they can take into account interactions between soil properties. Therefore, the importance of SOC likely varies depending on other local controlling factors such as clay content or climate or even the nature of the organic compounds, which could explain the strong effect of N. In other words, ML-PTFs are able to partially compensate for the effect of using a fixed conversion factor between SOC and SOM."

Apologises for this long reply to your suggestion. Hope you are satisfied with it.

**References in the manuscript**

Pribyl, D.W.: A critical review of the conventional SOC to SOM conversion factor, Geoderma, 156(3–4), 75–83, https://doi.org/10.1016/j.geoderma.2010.02.003, 2010.

Sprengel, C.: Ueber Pflanzenhumus, Humussaüre und humussaure Salze, Archiv für die Gesammte Naturlehre, 8, 145-220, 1826.

Van Bemmelen, J.M.: Über die Bestimmung des Wassers, des Humus, des

Schwefels, der in den colloïdalen Silikaten gebundenen Kieselsäure, des Mangans u. s. w. im Ackerboden Die Landwirthschaftlichen Versuchs-Stationen, 37, 279-290, 1890.

**References in the responses**

Adams, W.A.: The effect of organic matter on the bulk and true densities of some uncultivated podzolic soils, J. Soil Sci., 24(1), 10-17, 1973.

Jolivet, C., Arrouays, D., and Bernoux, M.: Comparison between analytical methods for organic carbon and organic matter determination in sandy Spodosols of France, Commun. Soil Sci. Plant Anal., 29(15-16), 2227-2233, https://doi.org/10.1080/00103629809370106, 1998.

Mann, L.K.: Changes in soil carbon storage after cultivation, Soil Sci. 142, 279–288, 1986.

Pribyl, D.W.: A critical review of the conventional SOC to SOM conversion factor, Geoderma, 156(3–4), 75–83, https://doi.org/10.1016/j.geoderma.2010.02.003, 2010.

Sprengel, C.: Ueber Pflanzenhumus, Humussaüre und humussaure Salze, Archiv für die Gesammte Naturlehre, 8, 145-220, 1826.

Sun, W., Canadell, J. G., Yu, L., Yu, L., Zhang, W., Smith, P., Fischer, T., and Huang, Y.: Climate drives global soil carbon sequestration and crop yield changes under conservation agriculture, Glob. Chang. Biol., 26, 3325-3335, https://doi.org/10.1111/gcb.15001, 2020.

Van Bemmelen, J.M.: Über die Bestimmung des Wassers, des Humus, des Schwefels, der in den colloïdalen Silikaten gebundenen Kieselsäure, des Mangans u. s. w. im Ackerboden Die Landwirthschaftlichen Versuchs-Stationen, 37, 279-290, 1890.

2. Please explain a bit more about the depth of the soil samples. It seems that you only build the models and compare your models with other models for the depth of 0-20 cm. My question here is that your machine learning models share the same climate/terrain predictors but different soil property predictors. What

will happen if the authors want to estimate soil bulk density at a soil profile at different depths or even for 0-5 cm or 0-15 cm within the dataset you have? Will the coefficients stay the same for different depths? It may be helpful to publish another reduced model without climate/terrain predictors for an improved applicability/transferability of your more accurate models so that researchers can use them for bulk density estimation at different depths, just like the current models.

Response: Thanks for your simulating comments and suggestions. The LUCAS Soil only recorded soil information at 0-20 cm deep for the last three sampling campaigns (for the year of 2009, 2015 and 2018). Therefore, we have only generated all the PTFs for a depth at 0-20 cm and we are not able to generate a model applicable for deeper soil layers or upper ones (0-5 cm). However, we stress in the paper that this should be an objective for the future. This limitation has been mentioned in the discussion in Lines 358-363: "It is essential to acknowledge that our developed PTFs for $BD_{fine}$ prediction was constructed based on LUCAS Soil data (0-20 cm), confining its applicability to topsoil within the EU and UK (Orgiazzi et al., 2022, Panagos et al., 2022). However, the potential of their extrapolation capability to other regions or to deep soil (>20 cm) necessitates further evaluation. As more soil data become available from diverse regions as well as for deep soil (Lal, 2018; Tautges et al., 2019; Batjes et al., 2020; Yost et al., 2020), the proposed methodology can be further used to update the PTFs, thereby broadening its area of applicability (Chen et al., 2018; Meyer and Pebesma, 2021).".

Of course, if people have access to depth-specific soil bulk density data, they can develop depth-explicit machine learning models to account for the effects of climate and terrain on bulk density at depths. However, I think the research community has not well studied this issue and it will be a very important topic for future collaboration.

Response: Your suggestion is quite simulating for future work. We share the

same point of view that depth-explicit machine learning based PTFs would be important for more accurate estimate BD since it can account for the effects of climate and terrain on bulk density at depths. As you mentioned, depth-specific soil bulk density data along with depth-specific soil properties would be required to build such PTFs. This suggested has been added in Lines 363-365: "In addition, when a depth-specific soil $BD_{fine}$ database is available, it will be important to develop depth-explicit ML-PTFs to account for the effects of climate and terrain on $BD_{fine}$ at depths.". In summary, we fully agree with you that this will be a very important topic for future collaboration.

---

## Author Comment (AC3)

**Responses to Reviewer 3**

Please note that your comments are provided in green text and our responses are marked in blue text. Our major modifications in the revised manuscript are marked as red text.

This paper addresses the question of predicting the bulk density of soils, its absence in soil databases limiting our ability to move from soil mass characteristics (quantities per unit mass of soil) to characteristics expressed in relation to a volume of soil or to a surface area of soil for a given soil thickness. This is an extremely important subject. The soil databases have bulk density values for a minority of soils stored there, but enough to make the study possible of how it is possible to predict, using pedotransfer functions (PTFs), the bulk density using other characteristics of these soils for which the bulk density values are available. The objective is to have tools for predicting the bulk density using soil characteristics that are much more easily accessible than the bulk density. Here, the measured and predicted values of bulk density are then used to compute the stock of soil organic carbon. The latter are discussed according to the characteristics of the PTFs used and the characteristics of the soils, including their environmental characteristics. This is an article which deserves to be published in "Earth System Science Data" but which must first be corrected both in substance and form according to the comments which follow.

Response: Many thanks for your positive feedbacks as well as suggestive comments on our manuscript. We have carefully revised the manuscript based on your comments and suggestions and we hope that the revised manuscript has been greatly improved thanks to your help. Please find our point-to-point responses to your concerns below.

I did not find a presentation of the way used to discuss the "accuracy" of the prediction of the bulk density and then of the soil organic carbon (SOC) content. This requires to be improved (see also comments along the text). "Accuracy" is discussed using the R2 and RMQS values alone. I recommend going deeper in this area. This should be a major point of the discussion.

Response: Thanks for your helpful comments. The use of "accuracy" is confusing and therefore we have replaced it by "model performance" in the revised manuscript. In addition to $R^2$ and RMSE, we have added relative error (RE) as you suggested for the evaluation of model performance (in Lines 185-190). In addition, we have added relevant discussion on the model performance of PTFs under different $BD_{fine}$ levels and land covers to provide more insightful information for the readers. Please find detailed responses to these specific comments below. Hope you are satisfied with our revision.

There are a certain number of assertions in the discussion section: "better choice for improving BD prediction" (better than what?) (Line 240); "can be an efficient tool" (To what respect?) (Line 247), "greatly improved" (improved but not greatly) (Line 250); "performed better" (this should be more appropriately discussed) (Line 278); "would be accurate enough" (enough with respect to what consideration?) (Line 282). Such assertions that are not clearly supported by facts cannot be accepted.

Response: Thanks for your detailed comments regarding our statements. We have revised all the relevant assertions to make our statements more objective and to avoid confusions.

Former Line 240, new Lines 293-294: "a better choice for improving BD prediction than earlier published PTFs based on algebraic equations.".

Former Line 247, new Lines 301-303: (instead of "can be an efficient tool"). "The comparison between global PTFs and local PTFs performances shows that local PTFs can improve the efficiency for imputing missing data using a large soil database (Padarian et al., 2019; Sanderman et al., 2020).".

Former Line 250, new Lines 304: We wrote "substantially" instead of "greatly".

Former Line 278, new Lines 346-348: We feel that the demonstration by the scatter plots is clear enough. We reformulated as "Looking into the scatter plots shown in Fig. 5, we can observe that the ML-PTFs performed much better than earlier published PTFs for topsoil samples with high BD (and low SOC content) while limited difference was found for soil samples with low BD (and high SOC content).".

Former Line 282: "would be accurate enough", new Lines 350-354: We reformulated the demonstration as following "As a result, the best earlier published PTF (PTF-4) performed quite similar to the best ML-PTF (local-RF$_{FRFS}$) when considering the topsoil samples with a wide range of SOC stock. This last result suggests that earlier published PTFs could be useful default tools to estimate BD which is subsequently used for SOC stock calculation. One of the advantages of these earlier published PTFs is their simplicity; another obvious advantage is that they require less numerous learning points than ML-PTFs to be fitted and validated.".

The authors do not use always the same abbreviation for the bulk density and the different pedotransfer functions (see comments along the text). There are also other abbreviations which vary in the text (see also comments along the text). This does not make easy reading and understanding the text. Please homogenize all the abbreviations throughout the whole text.

Response: Thank you for this important comment. We have carefully checked and homogenized all the abbreviations throughout the whole text in the revised manuscript.

There are several (too many) writing errors which reflect a lack of proofreading of the manuscript before submitting it. There is even an equation that is wrong in the text even though the calculations appear to have been carried out correctly. (Eq. 3, Line 174). There are enough co-authors to take care of this

proofreading work. Please see comments along the text. It is not pleasant for reviewer's work.

Response: We are sorry and we apologise for leaving so many writing errors in the original manuscript. The revised manuscript has been carefully checked regarding the grammars and equations. Thank you very much for your patience and your help in pointing out all the relevant issues below.

Legends of Figures and Tables require to be much more informative.

Response: Thanks for your suggestion. We have added more descriptions in the captions of figures and tables. They are more informative in the revised manuscript.

Title: The discussion is restricted to the discussion of the topsoil bulk density (i.e. 0-20 cm). The question of the prediction of the bulk density concerns both the topsoil and subsoil horizons. I have no problem with focusing the prediction on the topsoil when the objective is predicting the soil organic carbon content because the stock is mainly located in the topsoil horizons. However, this should be indicated more explicitly in the title by using "topsoil bulk density" instead of "soil bulk density". Then, I am wondering about the singular form for "pedotransfer function". It would be more appropriate to use the plural form "pedotransfer functions".

Response: Thanks for pointing out this important issue. Since the LUCAS Soil database only collected data at topsoil (0-20 cm), the PTFs fitted in this manuscript were restricted to topsoil in both the results and discussion. Indeed, we have mentioned the limitation of our study on only focusing on topsoil as well as the importance for building depth-specific ML-PTFs for BD when more soil profiles data is available (in Lines 358-365). We agree with your suggestion on the title, and we revised it as "European topsoil bulk density and organic carbon stock database (0-20 cm) using machine learning based pedotransfer functions". Hope you are happy with our revision.

Line 35: "Additionally, BD plays a crucial role in calculating SOC storage" I recommend starting with a sentence more general like "Additionally, BD plays a crucial role in computing stock of water, chemical elements or compounds by soil surface unit or soil volume unit and then focusing on SOC stocks.

Response: Thanks for your nice suggestion. We have revised it as your suggested in Lines 37-39: "Additionally, BD plays a crucial role in computing stock of water, chemical elements (e.g., soil organic carbon, SOC) or compounds by soil surface unit or soil volume unit, making it even more essential in soil studies.".

Lines 38 & 39: "to acknowledge … cover patterns" This sentence is correct if you are speaking about the topsoil bulk density. For the subsoil bulk density, the latter closely varies according to soil texture. Please the authors should restrict to topsoils.

Response: We have restricted this statement to topsoil (in Line 41) as you suggested.

Line 46: SOC is soil organic carbon content. Please the authors should add "content" to "SOC" and also to "clay, silt, sand" everywhere in the whole text.

Response: Thanks for your suggestion. We have added "content" after SOC, clay, silt, sand throughout the manuscript.

Lines 79 & 80: "data under comparable environmental conditions" Very vague. Please, it is required to be more specific.

Response: Similar environmental conditions would be more appropriate here and we have revised it accordingly in Lines 83-84 and detailed what "environmental conditions" means in this context as below "similar environmental conditions (i.e. in the present case similar predictors feature space, including soil properties, elevation, land cover and climate conditions).".

Line 83: "accuracy" What is "accuracy" in this paper. How is it expressed, discussed? See other comments about that point.

Response: You're right, accuracy is a vague term including many aspects. We prefer the generic term "performance". This wording is commonly accepted for ML predictions and includes indicators as $R^2$, root mean square error, relative error, and other indicators. We rewrote as in Lines 87-88: "…how the performances (e.g., $R^2$, root mean square error, relative error) of PTFs based BD prediction impact the quality of SOC stock remains poorly explored.".

Line 96: All throughout the text the word "soil" is used when it is the "topsoil" (0-20 cm) which is discussed. It is necessary to avoid such an ambiguity.

Response: Thanks for this suggestion. We replaced soil by topsoil in all the relevant positions.

Line 99: What do the authors mean by "a single laboratory". If the analyses were performed in a single laboratory, please give information about this laboratory.

Response: Thanks for your suggestion, we have added the relevant information in Line 104: "Standard laboratory analysis was conducted in an accredited laboratory (Kecskemét, Hungary),".

Lines 100 & 100: "-3" and "−1" require to be written in superscript.

Response: Thanks for pointing out these typos. All the relevant typos have been corrected in the revised manuscript.

Line 120: "Traditional" Is it appropriate? I do not think so. I would suggest using "Earlier published PTFs" or "PTFs from the literature". I do not understand in why these PTFs would be "traditional". And what tradition are we talking about? Unclear and not adapted.

Response: Thanks for this nice suggestion. We agree with your that "Earlier published PTFs" would be clearer and we have revised it accordingly throughout the manuscript.

Line 125: In table 2, four models are presented and numbered 1, 2, 3 and 4 when there are mentioned as PFT-1, PFT-2, PFT-3 and PFT-4 in Figure 5 (Line 207) and Figure 6 (Line 2014). I mention here that the correct abbreviation for "pedotransfer function" is "PTF" and not "PFT" as mentioned in Figures 5 and 6. What is BD in Table 2? BDfine? SOM content is defined as % by reference to soil mass or soil volume? Same question for the SOC content.

Response: Thanks for your great patience. We have named the earlier published PTFs as PTF-1, PTF-2, PTF-3 and PTF-4 in Table 2. The same abbreviations have been used in the whole revised manuscript. We have also corrected the typo of "PFT" in Figure 5. The $BD_{fine}$ and $CF_{volumefraction}$ have been carefully used in the revised manuscript. SOM and SOC contents are defined as % by reference to soil mass. This information has been added in Table 2.

Line 134: "Here, 16 predictor variables" when there are 15 predictors mentioned in Table 3. Please check.

Response: Sorry for this mistake. We have corrected it as 15 predictor variables in Line 147.

Line 135: Table 3. I do not understand using the three clay, silt and sand contents together (RFFull) when they are not independent predictor variables, their sum being equal to 100. For RFRFS, sand content is not used. Please explain. "Elevation" in the table when it is "DEM" in the text (Line 116). Please homogenize. "EC" in the table when it is "CEC" in the text (Line 101)/ Please homogenize. "ELE" for probably "elevation" when it is not defined in the text. This is confusing.

Response: We added this text to the caption of Table 3: "$RF_{Full}$ uses all potential

predictors, even if they may be redundant of multi-collinear (typical case of the use of clay, silt and sand contents together). RF$_{FRFS}$ applies forward recursive feature selection thus eliminating both multi-collinearity and irrelevant covariates (e.g., one particle size fraction (sand) is left out).". Please note the fact that sand is out is linked to the fact that clay and sand generally have the strongest negative correlation because they are at the extreme of the textural triangle whereas silt usually has weaker correlations with other fractions. This is also understandable from a sedimentology point of view. The abbreviation of elevation (ELE) has been defined in Line 126. EC has been corrected as CEC in Table 3. All the abbreviations have been carefully checked in the revised manuscript.

Lines 136 & 137: "Furthermore, we adopted … performance". Please give at least one reference.

Response: We have added two relevant references in Lines 157-158 as you suggested.

Lines 160 to 167: I recommend discussing errors using relative errors. Is the error 5%, 10%, 15% or more of the predicted value? Is there any relationship between the relative error and the type of land use? The discussion of the prediction quality would be thus much more relevant.

Response: We have added relative error (RE) as one of the indicators for evaluating the performance of PTFs (in Lines 185-190). We have added RE in the Figure 5 and Figure 7 (former Figure 6), and also added a new Figure 6 to demonstrate the RE variations among different BD levels and land covers. The results showed that the magnitude of RE depended on BD$_{fine}$ levels in Lines 236-243: "The summary of RE variations under different BD$_{fine}$ levels and land covers using best earlier published PTF (PTF-4) and ML-PTF (local-RF$_{FRFS}$) is shown in Fig. 6. The results indicated that local-RF$_{FRFS}$ (RE of 29%) performed much better than PTF-4 (RE of 37%) for the topsoil with low BD$_{fine}$ (<0.8 g cm$^-$

[3]). The improvement of RE for other $BD_{fine}$ levels was rather limited ($\Delta RE$ of 1-3%). The highest RE (30-57% for PTF-4, 25-50% for local-$RF_{FRFS}$) was found for topsoil with low $BD_{fine}$ for the whole validation set and each land cover. Across land covers, the RE generally decreased greatly (15-24% for PTF-4, 14-20% for local-$RF_{FRFS}$) for topsoil with low-median $BD_{fine}$ (0.8-1 g cm$^{-3}$), and then to its lowest (7-9% for both PTF-4 and local-$RF_{FRFS}$) for topsoil with median-high BD (1-1.2 g cm$^{-3}$). A slight increase of RE (14-16% for PTF-4, 11-17% for local-$RF_{FRFS}$) was observed for topsoil with high $BD_{fine}$ (>1.2 g cm$^{-3}$) for all the land covers.".

RE also varied a lot among land covers for topsoil with low and low-median $BD_{fine}$ in Lines 243-247: "Among different land covers, the cropland had the greatest RE for topsoil with low and low-median $BD_{fine}$, followed by others, woodland and grassland. For topsoil with median-high and high $BD_{fine}$, similar RE was found for all the land covers. Overall, the RE both PTF-4 and local-$RF_{FRFS}$ showed the worse performances for low $BD_{fine}$ values, but the results were always better for local-$RF_{FRFS}$, except for woodlands having $BD_{fine}$>1 where the RE was slightly better for PTF-4".

We have also added relevant discussions on RE in Lines 314-325: "Looking into the RE for topsoil under different $BD_{fine}$ levels (Fig. 6), it is clear that the fitted best PTFs (PTF-4 and local-$RF_{FRFS}$) had the highest REs for topsoil with low $BD_{fine}$ (<0.8 g cm$^{-3}$) despite that local-$RF_{FRFS}$ performed better. This partly results from the low $BD_{fine}$ to calculate the RE, because $BD_{fine}$ value is used as the reference 100% value in RE calculation. This is also likely due to the general trend of broad-scale predictions to smooth the variability and to overestimate the lowest values and to underestimate the higher values whatever the predicted variable is (e.g., Tifafi et al., 2018; Lemercier et al. 2022; Richer-de-Forges et al., 2023). Most important, many low $BD_{fine}$ observed values are probably linked to large voids resulting in a large porosity, especially under disturbed topsoils. This explains why cropland topsoils exhibited such a large RE, likely due to the effect of soil tillage which cannot be predicted by our

covariates. This can also explain the decreasing trend of RE with the increase of $BD_{fine}$ up to 1.2 g cm$^{-3}$ whereas for the topsoil with high BD (>1.2 g cm$^{-3}$), both local-RF$_{FRFS}$ and PTF-4 showed a slight increase in RE. Overall, the RE might appear a bit deceiving if we compare them to the precision that one may wish for monitoring changes in $BD_{fine}$ for example as an indicator of compaction. We must state that this is clearly out of the scope of this study, which is to provide a wide database of reference values that can be used for broad-scale modelling.".

Line 168: Is it BD or BDfine? Same question for Line 177 and Figure 2. This really confusing.

Response: Thanks for your comment. We have specified the $BD_{fine}$ here as well as in other texts, equations, figures and tables.

Line 174: Equation (3) appears to be wrong. How is expressed CFvolumefarction? Does it range from 0 to 1? From 0 to100? It should be "x (1 - CFvolumefarction)" without dividing by 100 if CFvolumefarction ranges from 0 to 1 or "x (100 - CFvolumefarction)/100" if CFvolumefarction ranges from 0 to 100. Required to be clarified and corrected.

Response: The CF ranges from 0 to 1, the unit of CF is %/100 which has been specified in Line 198. The 100 at the end of the equation 3 is not used to correct the unit of CF, but for converting the final unit of SOC stock to kg m$^{-2}$. Hope our explanation is clear. Please note that this equation (new Equation 4) have been revised for better understanding.

Line 178: "with BD ranging from 0.20 to 1.89". This required to be discussed in the discussion section. For which type of topsoil do we encounter 0.20? Peat topsoils? And for 1.89? Stony topsoils? But are we talking about BDfine or BD including gravels and stones? This remains confusing.

Response: Thanks for your suggestions. We have added the descriptions on

the lowest and highest BD$_{fine}$ in Lines 203-206: "The topsoil sample with the lowest BD$_{fine}$ (0.20 g cm$^{-3}$) was collected from Pine dominated mixed woodland with a SOC content greater than 137 g kg$^{-1}$. In contrast, the topsoil sample with the highest BD$_{fine}$ (1.89 g cm$^{-3}$) was sampled at a sandy soil (sand and clay of 65% and 11%, SOC content of 31.9 g kg$^{-1}$) in cropland (common wheat).". Hope our explanations are clear.

Line 180: "with the exception of clay soils". First of all, you are talking about "topsoils" and not "soils" and then there are clayey topsoils in your dataset (see the triangle, Figure 2) and not so few. This requires to be rewritten.

Response: You are right. Indeed, the textural triangle is well covered. Thanks for your suggestion. We have restricted it to topsoil now and we have rewritten the sentences in Lines 207-208 as "As shown in the texture triangle, the selected topsoil samples covered a wide range of soil texture classes.".

Line 193: "Elevation" here when it is for RFFRFS in Table 3. Please homogenize.

Response: Elevation has been replaced by ELE in relevant positions.

Lines 196 to 203 (and elsewhere in the text, Figures 5 and 6 included): The abbreviations ML-PTFs and T-PTFs are used in the text which is appropriate. I strongly suggest using local-RFFRFS-PTFs, local-RF-FULL-PTFs and so on for the other PTFs to homogenize and make easier text reading and understanding.

Response: Based on your previous suggestion, the earlier published PTFs have been named from PTF-1 to PTF-4, and the ML-PTFs have been named as global-RF$_{FULL}$, global-RF$_{FRFS}$, local-RF$_{FULL}$ and local-RF$_{FRFS}$. Hope the current version is easier for reading and understanding.

Line 205: The legend is not informative enough. Please avoid mentioning "eight PTFs". This does not bring any information.

Response: Thanks for this suggestion. The caption of Figure 4 has been revised as "Figure 4 Model performance indicator ($R^2$) of earlier published PTFs and ML-PTFs in $BD_{fine}$ prediction. The performances of local RF models (local-$RF_{FULL}$ and local-$RF_{FRFS}$) change with the number of soil samples used for local modelling.". Hope this caption is more informative.

Line 207: Figure 5. As mentioned above, this not "PFT" but "PTF". The legend of the figure is not informative enough. Please avoid mentioning "eight PTFs". This does not bring any information.

Response: We have corrected the typo of "PFT" as "PTF" in Figure 5. The caption of Figure 5 has been revised as "Figure 5 Scatter plots of $BD_{fine}$ predictions using earlier published PTFs and ML-PTFs along with model performance indicators (RMSE, $R^2$ and RE). The lighter color means higher sample density. Please note that the best models are selected for local-$RF_{FULL}$ and local-$RF_{FRFS}$.". Hope this caption is much informative.

Line 214: Figure 6. Similar comments as in Figure 5. SOC stocks are expressed in kg cm-2 which is wrong. Probably should correspond to kg m-2. When the authors write "observed SOC stocks", I assume that they are speaking about values of SOC stocks which were computed using the measured values of SOC content and measured values of bulk density. And then, when they write "Predicted so stocks", the values were computed using measured values of SOC content and the predicted values of bulk density. Whether I understood correctly or not, it is necessary to explain it clearly in the text.

Response: Sorry for this mistake, the unit of SOC stocks has been corrected as "kg m$^{-2}$". Indeed, your understanding is correct and the relevant descriptions have been added in the caption of Figure 7 (former Figure 6): "Figure 7 Scatter plots of SOC stock predictions by earlier published PTFs and ML-PTFs along with model performance indicators (RMSE, $R^2$ and RE). The red points represent topsoil samples with SOC stock<3 kg m$^{-2}$ while the blue points

represent topsoil samples with SOC stock≥3 kg m$^{-2}$. Please note that observed SOC stock is computed using SOC content, $CF_{volumefraction}$, $BD_{fine}$ observations, and while predicted SOC stock is computed using SOC content, $BD_{fine}$ predictions and $CF_{volumefraction}$ transformed from $CF_{massfraction}$ using $BD_{fine}$ predictions suggested by Pacini et al. (2023).".

Lines 249 to 252: This is not really true. The difference of R2 is not "around 2.0". The highest difference of R2 recorded with T-PTFs and with the PTFs developed in the paper is 0.19 when we compare the smallest R2 recorded with T-PTFs and the highest R2 recorded with the PTFs developed in the paper (see values in Figure 5). On the other hand, the difference of R2 recorded with T-PTFs and with the PTFs developed in the paper is 0.14 when we compare the highest R2 recorded with T-PTFs and the highest R2 recorded with the PTFs developed in the paper (see values in Figure 5). I recommend writhing something like "ranged from 0.14 to 0.19" which more appropriate.

Response: Many thanks for your kind suggestion. We agree with you that "$\Delta R^2$ of 0.14-0.19" would be more appropriate, and we have revised it in Line 305 in the revised manuscript.

Line 247: "can be an efficient tool" Meaning? Something with "can improve" would much more appropriate.

Response: Thanks for this suggestion. We have revised it in Lines 301-303 as "Therefore, the comparison between global PTFS and local PTFs performances shows that local PTFs can improve the efficiency for imputing missing data using a large soil database (Padarian et al., 2019; Sanderman et al., 2020).".

Lines 270 & 271: "with a higher SOC commonly". "with a higher SOC content commonly" is more correct. And "higher" than what? "larger" than what? "greater" than what? The use of the comparative form requires to say to what

you compare.

Response: Thanks for your suggestions. We have added content after SOC in the whole manuscript to make it clear. To avoid confusion, we have revised the sentences not to make them "comparative" in Lines 338-340: "For instance, a soil sample with a high SOC content commonly has a large pore space due to the large amount of organic matter, leading to a low $BD_{fine}$ (Perie and Ouimet, 2008; Chen et al., 2018).".

Line 280 ">3 kg cm-2" Quite high. I assume this is "3 kg m-2"

Response: Thanks for pointing out this typo and we have corrected it in the revised manuscript. Similar typos also have been carefully corrected in the whole manuscript.

Line 280: "would be accurate enough" Why? Based on what? This requires to be clarified.

Response: We agree with your comment that our statement is confusing here. We have revised it in Lines 352-353: "This last result suggests that earlier published PTFs could be useful default tools to estimate $BD_{fine}$ which is subsequently used for SOC stock calculation.". Hope this statement is clear now.

Line 280: "to topsoil" This is the only place where we are talking about topsoils and not soils.

Response: Many thanks for your previous relevant suggestions. We have replaced soil by topsoil in the revised manuscript to avoid confusion.

I did not check that all the references cited in the text were in the reference list and vice versa.

Response: Thanks again for all your helpful comments and suggestions above. We have carefully checked the references in the revised manuscript.

---

## Author Comment (AC4)

**Responses to Reviewer 4**

Please note that your comments are provided in green text and our responses are marked in blue text. Our major modifications in the revised manuscript are marked as red text.

The authors proposed an interesting topic that addresses the need for the availability of reliable data on soil properties that are crucial for many assessments of soil quality indicators. The authors, in addition to evaluating the performance in terms of accuracy of traditional PTFs and of four proposed machine learning (ML) based PTFs, assessed the impact of their accuracy on that of the estimated SOC stock. This is a very qualifying point of the manuscript in which a problem rarely considered is addressed. Indeed, neglecting the accuracy of input data in estimating soil carbon stock is a major problem that can lead to under- or over-estimation.

Response: We highly appreciate your positive feedbacks on our work. We fully agree with your point of view that the assessing the impact of BD accuracy from PTFs on the estimated SOC stock. This kind of assessment can provide a reference for evaluating the uncertainty propagation of PTFs on other derived soil properties, enabling a more reasonable use of PTFs outputs. Thanks again for your nice summary on our work.

The manuscript is well organised and clear with a sound application of the methods used and it is not easy to find flaws beyond the few minor ones that have been pointed out by other reviewers.

Response: Thanks for your kind comments. We have carefully revised all the issues suggested by other three reviewers, and we hope the quality of the revised manuscript has been greatly improved.